# Ferromagnets, a new anomaly, instantons, and (noninvertible) continuous translations

Nathan Seiberg

School of Natural Sciences, Institute for Advanced Study, Princeton, NJ, USA

## Abstract

We discuss a large class of classical field theories with continuous translation symmetry. In the quantum theory, a new anomaly explicitly breaks this translation symmetry to a discrete symmetry. Furthermore, this discrete translation symmetry is extended by a $d-2$-form global symmetry. All these theories can be described as $U(1)$ gauge theories where Gauss law states that the system has nonzero charge density. Special cases of such systems can be phrased as theories with a compact phase space. Examples are ferromagnets and lattices in the lowest Landau level. In some cases, the broken continuous translation symmetry can be resurrected as a noninvertible symmetry. We clarify the relation between the discrete translation symmetry of the continuum theory and the discrete translation symmetry of an underlying lattice model. Our treatment unifies, clarifies, and extends earlier works on the same subject.

# 1 Introduction

It is almost always the case that a lattice theory is described at long distances by a continuum field theory. Even though the underlying lattice model has only discrete translation symmetry, the continuum theory is invariant under continuous translations. However, a number of examples, e.g., the continuum theories of ferromagnets and quantum crystals in the lowest Landau level[1] [5–16] have defied that expectation. As we will discuss, these phenomena are related to another subtle effect, the breaking of translation symmetry in continuum QED at finite density [17–19].

The goal of this paper is to present a unified treatment of many continuum quantum field theories that are classically translation invariant, but due to a new quantum anomaly, the continuous translations symmetry is explicitly broken to discrete translations. This is the case even though space is still continuous. Furthermore, these discrete translations do not commute.

All these theories can be described as $U(1)$ gauge theories with a classical Lagrangian density of the form[2]

$$\mathcal{L}^{Classical} = \frac{k}{V}a_t + \mathcal{L}^{(0)}. \tag{1}$$

---

[1]See, e.g., the textbooks [1–4], for discussion of these and related systems.

[2]Throughout this note we will study theories in $d$ spatial dimensions. Our notation is that Lorentzian signature time is denoted by $t$ and Euclidean signature time is denoted by $\tau$. The spatial indices are denoted by $i, j, \ldots$ and spacetime indices are denoted by $\mu = t, i, j, \ldots$ or $\mu = \tau, i, j, \ldots$. Also, we will take space to be a $d$-dimensional torus $\mathbb{T}^d$ parameterized by $x^i \sim x^i + \ell^i$. Our conventions are such that we contract the spatial indices with $\delta_{ij}$. This means that we do not distinguish between upper and lower spatial indices.

Here, $a_\mu$ is the $U(1)$ gauge field and

$$V = \prod_i \ell^i, \tag{2}$$

is the total spatial volume. Classically the dimensionless coefficient $k$ is arbitrary, but in the quantum theory, $k$ has to be an integer.[3] $\mathcal{L}^{(0)}$ includes kinetic terms and various interaction terms of all the fields in the problem. These terms are gauge invariant and are manifestly translation invariant.

The superscript *Classical* in (1) means that this Lagrangian density can be used to find the classical equations of motion. Such Lagrangian densities are not always well-defined, but the equations of motion derived from them are meaningful. In the quantum theory, we can still have ill-defined Lagrangian density $\mathcal{L}$, but the integrand in the functional integral $\exp\left(i \int d^d x\, dt\, \mathcal{L}\right)$, or its Euclidean version $\exp\left(-\int d^d x\, d\tau\, \mathcal{L}_{Euclidean}\right)$, should be meaningful. In the course of defining it, one might need to add to $\mathcal{L}^{Classical}$ "correction terms" that do not contribute to the classical equations of motion.[4]

Classically, all the first term in (1) does is to shift Gauss law by a constant $\frac{k}{V}$ representing fixed charged density. We will study its effects in the quantum theory. Specifically, the careful definition of this term will involve choosing a reference point in space $x_0^i$, thus explicitly breaking the naive continuous translation symmetry. More explicitly, under translations,

$$x^i \to x^i + \epsilon^i, \tag{3}$$

the properly defined quantum Lagrangian density $\mathcal{L}$ transforms as

$$\mathcal{L} \to \mathcal{L} + \epsilon^i \frac{k}{V} f_{it},$$
$$f_{\mu\nu} = \partial_\mu a_\nu - \partial_\nu a_\mu. \tag{4}$$

We will interpret this phenomenon as analogous to the Adler-Bell-Jackiw chiral anomaly, except that it is between the $U(1)$ gauge symmetry and the translation symmetry.[5]

As we will see, instantons, i.e., Euclidean spacetime configurations with nonzero

$$\mathcal{Q}_{i\tau} = \frac{1}{2\pi} \int dx^i d\tau\, f_{i\tau} \in \mathbb{Z}, \tag{5}$$

activate this anomaly and break the continuous $U(1)^d$ translation symmetry to discrete subgroup $\mathbb{Z}_k^d$. Furthermore, that discrete symmetry is extended by the $d-2$-form magnetic global symmetry of the $U(1)$ gauge field [20] and becomes non-Abelian.

Special cases of this result were discussed in [17–19] and others arise in various models based on a local compact phase space $\mathcal{P}$. (Below, we will review why this is a special case of (1).) In this context, the fields are local coordinates $\phi^r$ on $\mathcal{P}$. The phase space is characterized by a symplectic structure $\mathcal{F}(\phi)_{rs} = \partial_r \mathcal{A}_s - \partial_s \mathcal{A}_r$, where the Liouville form $\mathcal{A}_r$ is not globally well-defined. Then, we study theories based on classical Lagrangian densities of the form

$$\mathcal{L}^{Classical} = \frac{k}{V} \sum_r \mathcal{A}_r \partial_t \phi^r + \mathcal{L}^{(0)}. \tag{6}$$

---

[3]More precisely, we set $\hbar = 1$. Then, the classical limit corresponds to $k \to \infty$ with fixed $V$ and fixed $\frac{1}{k}\mathcal{L}^{(0)}$.

[4]All our continuum theories are non-renormalizable and should be viewed as effective theories.

[5]An 't Hooft anomaly is an obstruction to gauging. The Adler-Bell-Jackiw (ABJ) anomaly arises when we gauge an anomaly-free symmetry, but due to the underlying 't Hooft anomaly in the problem, another symmetry is broken. In the original case, studied by Adler, Bell, and Jackiw, the latter was a chiral symmetry. In our case, it is the translation symmetry. We will return to 't Hooft anomalies in Section 6.3.

The first term is often referred to as a Berry term or as a Wess-Zumino term. Classically, it adds to the equations of motion of $\phi^s$ the well-defined term $\frac{k}{V} \sum_r \mathcal{F}_{sr} \partial_t \phi^r$. And as above, $\mathcal{L}^{(0)}$ includes various other terms, all of which are globally well defined.

As in the more general discussion of the $U(1)$ gauge theory (1), in the quantum theory, (6) should be defined carefully. Again, this will uncover an anomaly, which is activated by instantons. It breaks the translation symmetry to a discrete subgroup and extends it to a non-Abelian group.

Many people have studied such Lagrangians in various contexts and have found closely related facts. In particular, [17] and later [18, 19] have studied QED at finite density, which is described by a Lagrangian of the form (1) and found the breaking of translation symmetry. Our discussion will be similar to that of [18, 19]. The authors of [6, 7], followed by [8–10], and more recently, [13–15] have discussed the problems with translation symmetry of ferromagnets described by (6). Finally, in a different physical context, [21] pointed out that further data is needed to define the exponential of the action of (1) (but did not specify that extra data in detail). This issue was discussed further in [22].

One might try to study the translation symmetry of the Lagrangian densities (1) and (6) by following the standard relativistic Noether procedure expressions

$$\Theta^\nu_\mu = \left( \sum_r \frac{\partial \mathcal{L}}{\partial(\partial_\nu \phi^r)} \partial_\mu \phi^r \right) - \delta^\nu_\mu \mathcal{L}, \qquad \sum_\nu \partial_\nu \Theta^\nu_\mu = 0. \tag{7}$$

Using our non-relativistic notation, they are

$$\Theta_{tt} = \left( \sum_r \frac{\partial \mathcal{L}}{\partial(\partial_t \phi^r)} \partial_t \phi^r \right) - \mathcal{L}, \quad \Theta_{it} = -\sum_r \frac{\partial \mathcal{L}}{\partial(\partial_i \phi^r)} \partial_t \phi^r, \qquad \partial_t \Theta_{tt} = \sum_i \partial^i \Theta_{it}, \tag{8}$$

$$\Theta_{tj} = \sum_r \frac{\partial \mathcal{L}}{\partial(\partial_t \phi^r)} \partial_j \phi^r, \qquad \Theta_{ij} = -\left( \sum_r \frac{\partial \mathcal{L}}{\partial(\partial_i \phi^r)} \partial_j \phi^r \right) + \delta_{ij} \mathcal{L}, \quad \partial_t \Theta_{tj} = \sum_i \partial^i \Theta_{ij}.$$

However, the first term in (1) or the first term in (6) lead to ill-defined expressions for the momentum current $(\Theta_{tj}, \Theta_{ij})$.

Several authors have tried to address this issue with the momentum current and define a translation operator using various approaches. In the context of (6), one option depends on picking a reference point in the target space [6]. Another, follows Witten's description of the Wess-Zumino term [23] by adding another "bulk" dimension and expressing the operators as integrals over a larger space [6, 13, 14, 24]. (We will show that this is not always possible.) Some authors [8, 13] discussed an alternative momentum density current on infinite space. That current was interpreted in [16] as a dipole current. Finally, the lack of commutativity of translations in infinite volume was discussed in [12, 16]. In the context of the gauge theory (1), [18] modified the canonical momentum current to be gauge invariant, but not conserved and then used it to construct a discrete translation operator. That operator was later shown [19] to satisfy a noncommutative algebra.

Many of the elements in our discussion have appeared in these papers. But our treatment will differ from most of them in important ways:

- We will study the more general case (1) and later describe (6) as a special case that follows from it.

- We will discuss the theory in finite volume $V = \prod_i \ell^i$ with periodic boundary conditions. The reason is that the infinite volume limit is quite subtle and often singular. For example, if we want $k$ to be fixed, then the effect of the first term in (1) or (6) becomes

negligible in the limit. Alternatively, if we want the first term in (1) or (6) to have a nonzero effect, we should combine $V \to \infty$ with $k \to \infty$. Then, depending on how we scale $\ell^i$, the discrete $\mathbb{Z}_k$ translation symmetry in direction $j$ can become either $U(1)$ (if $\ell^j$ remains finite), or $\mathbb{Z}$ (if $\ell^j \to \infty$ with fixed $\ell^i$ with $i \neq j$), or $\mathbb{R}$ (if for at least one $i \neq j$, $\ell^j, \ell^i \to \infty$).

- We will view the continuum quantum field theory based on (1) or (6) as an effective theory whose classical limit has continuous translation symmetry. (The classical theory is as described in footnote 3.) Only later will we compare our continuum conclusions with an underlying lattice.

- In order to understand the origin of the translation symmetry breaking, we will focus on the theory, i.e., the Lagrangian densities (1) or (6), rather than on the details of momentum operator. This will lead us to conclude that the classical theory has continuous translation symmetry, while the quantum theory does not. In particular, a precise definition of the quantum theory depends on a choice of a reference point in space $x_0^i$, thus explicitly breaking the continuous translation symmetry. Then, the analysis of the translation operator will uncover additional structure.

It has recently been realized that some global symmetries that suffer from an Adler-Bell-Jackiw anomaly are resurrected as noninvertible symmetries. This was shown in [25, 26] for internal symmetries in the continuum and in [27, 28] for lattice translation. (See [29, 30], for reviews of noninvertible symmetries.) This motivated us to look for a similar phenomenon for continuum translations. Indeed, we will show that in some cases, the anomalous continuum translation that was explicitly broken by instantons, is resurrected as a noninvertible continuous translation symmetry.

This brings us back to the beginning of this introduction. These continuum models are the IR descriptions of UV lattice models, whose translation symmetry is discrete and Abelian. We will show that the lattice translation symmetry is an Abelian subgroup of the continuum non-Abelian discrete translation symmetry.

In Section 2, we will analyze the $U(1)$ gauge theory based on (1). We will define it carefully and will explore its translation symmetry. In Section 3, we will specialize to models with local phase space based on (6). We will relate them to the $U(1)$ gauge theories of Section 2 and will study their properties. In Section 4, we will resurrect the continuous translation symmetries of some of these continuum models as noninvertible translation symmetries. Section 5 will discuss the field theories of Section 2 and the special case in Section 3 as they arise from lattice models. Finally, in Section 6, we will summarize our results and will comment about extensions of these ideas.

## 2 Dynamical $U(1)$ gauge theories with background charge

In this section, we present a large class of $U(1)$ gauge theories that demonstrates our main point. The examples in Section 3 are special cases of these theories.

### 2.1 Comments about $U(1)$ gauge theory

We study a $U(1)$ gauge theory on a $D$-dimensional Euclidean torus parameterized by $x^\mu \sim x^\mu + \ell^\mu$ and denote the total volume by $\mathcal{V} = \prod_\mu \ell^\mu$. (In the applications below, this torus will be our $d$-dimensional space or our $d + 1$ dimensional Euclidean spacetime.) The gauge field is $a_\mu$ and the field strength is $f_{\mu\nu} = \partial_\mu a_\nu - \partial_\nu a_\mu$.

To define it carefully, we choose a local trivialization. We cover the torus with patches with transition functions between them. For simplicity, we take the overlaps to be along $D-1$-dimensional tori with fixed $x^\mu = x_*^\mu$.[6] The transition functions there are $\lambda^{(\mu)}$.

We would like to study integrals of the gauge fields. See [18] for a related discussion. First, we consider $\int dx^\mu a_\mu$ (no sum over $\mu$).[7] Since $a_\mu$ depends on the choice of transition functions, a better expression is $\int dx^\mu \left(a_\mu - \lambda^{(\mu)} \delta(x^\mu - x_*^\mu)\right)$, where we included a "correction term" due to the transition function at $x_*^\mu$. To make it fully gauge invariant, we exponentiate it to find the standard expression for the holonomy around the $\mu$-cycle

$$H_\mu = \exp\left(i \int dx^\mu \left(a_\mu - \lambda^{(\mu)} \delta(x^\mu - x_*^\mu)\right)\right). \tag{9}$$

Let us add another direction labeled by $\nu \neq \mu$ and try to study the integral $\int dx^\mu dx^\nu a_\mu$. This leads to two issues. First, the transition function $\lambda^{(\nu)}$ shifts $a_\mu$ at $x_*^\nu$ by $\partial_\mu \lambda^{(\nu)}$. Second, the lift of $\lambda^{(\mu)}$ to real numbers can jump at $x_*^\nu$ by $2\pi\mathbb{Z}$. As a result,

$$\frac{\partial}{\partial x_*^\nu} \int dx^\mu dx^\nu \left(a_\mu - \lambda^{(\mu)} \delta(x^\mu - x_*^\mu)\right) = 2\pi \mathcal{Q}_{\mu\nu} = \int dx^\mu dx^\nu f_{\mu\nu}. \tag{10}$$

The integer $\mathcal{Q}_{\mu\nu}$ is the first Chern-class and we expressed it using the magnetic flux through the $(\mu, \nu)$ cycle. One way to avoid this $x_*^\nu$ dependence is to choose a reference point $x_0^\nu$ and consider the integral

$$\int dx^\mu dx^\nu \left(a_\mu - \lambda^{(\mu)} \delta(x^\mu - x_*^\mu) + (x_*^\nu - x_0^\nu) f_{\mu\nu}\right). \tag{11}$$

Integrating over the other directions, this becomes

$$\int d^D x \left(a_\mu - \lambda^{(\mu)} \delta(x^\mu - x_*^\mu) + \sum_\nu (x_*^\nu - x_0^\nu) f_{\mu\nu}\right). \tag{12}$$

Finally, since $\lambda^{(\mu)}$ are circle-valued, the well-defined objects are

$$H_\mu^{(1,2,\ldots,D)} = \exp\left(\frac{i\ell^\mu}{\mathcal{V}} \int d^D x \left(a_\mu - \lambda^{(\mu)} \delta(x^\mu - x_*^\mu) + \sum_\nu (x_*^\nu - x_0^\nu) f_{\mu\nu}\right)\right). \tag{13}$$

Using this notation, the holonomy in (9) can be denoted as $H_\mu^{(\mu)}$.

We can phrase all this as follows. The logarithm of the holonomy around $x^\mu$ (9) is a circle-valued function of the other coordinates. We would like to integrate it over the other coordinates. To do that, we choose a local trivialization, i.e., choose $x_*^\nu$ and let it jump by $2\pi \mathcal{Q}_{\mu\nu}$ (with $\mathcal{Q}_{\mu\nu} \in \mathbb{Z}$) when we cross $x_*^\nu$. Clearly, the answer depends on $x_*^\nu$ (10). This dependence can be removed by adding the correction term in (11), or more generally (12).

To summarize, the expression $H_\mu^{(1,2,\ldots,D)}$ in (13) is gauge invariant and is independent of the choice of local trivialization including the values of $x_*^i$. However, it depends on our choice of $x_0^\mu$ and therefore it is not translation invariant.

---

[6]More precisely, the overlaps are these tori times a small segment around $x_*^\mu$.

[7]One might attempt to add another "bulk" dimension, parameterized by $u$, filling the circle parameterized by $x^\mu$. Then, extend $a_\mu$ to the bulk and replace $\int dx^\mu a_\mu$ by $\int dx^\mu du\, f_{u\mu}$. However, if the gauge field configuration is such that $\mathcal{Q}_{\mu\nu} = \frac{1}{2\pi} \int dx^\mu dx^\nu f_{\mu\nu} \neq 0$, there is no such smooth extension to the bulk. Indeed, below we will find interesting effects associated with nonzero $\mathcal{Q}_{\mu\nu}$.

## 2.2 The Lagrangian

### 2.2.1 Defining $\exp(i \int d^d x \mathcal{L}_a)$

We take space to be a $d$-dimensional torus $x^i \sim x^i + \ell^i$ with periodic boundary conditions. The dynamical fields include the $U(1)$ gauge field $a_\mu$ and various charged fields. The Lagrangian density is

$$\mathcal{L}_{U(1)}^{Classical} = \mathcal{L}_a + \mathcal{L}^{(0)}, \qquad \mathcal{L}_a = \frac{k}{V} a_t, \qquad V = \ell^1 \ell^2 \cdots \ell^d. \tag{14}$$

$\mathcal{L}^{(0)}$ describes the kinetic terms of the various fields and their interactions. Unlike $\mathcal{L}_a$, we take $\mathcal{L}^{(0)}$ to be locally gauge invariant. Also, we assume that it is manifestly translation invariant.

The main point is the unusual term $\mathcal{L}_a$ in (14). Such a term is common in quantum mechanical systems. Here, we will discuss some of its peculiar properties in field theory. Classically, it represents background charge density $\frac{k}{V}$ such that the equation of motion of $a_t$, i.e., Gauss law, states that the dynamical fields should screen this background charge.

$\mathcal{L}_a$ is gauge invariant, up to a total time derivative. Soon, we will use the discussion in Section 2.1 to define it carefully. For the time, we note that by compactifying Euclidean time $\tau \sim \tau + \beta$ and considering gauge transformations that wind around the compact Euclidean direction, we learn that the coefficient should be quantized

$$k \in \mathbb{Z}. \tag{15}$$

This is consistent with the interpretation of $k$ as the total background $U(1)$ charge.

As we emphasised in the Introduction, an unusual fact about the Lagrangian (14) is that it depends explicitly on the total volume $V$. This means that if we are interested in the

$$V \to \infty \tag{16}$$

limit, we can either take also $k \to \infty$ with fixed background charge per unit volume $\frac{k}{V}$, such that the coefficient has a nonzero limit, or we can take $V \to \infty$ with fixed $k$ and then the first term vanishes. Instead, we will be interested in the finite $V$ theory.

Since the first term in the Lagrangian is not locally gauge invariant, it should be defined carefully. We do it by going to compact Euclidean spacetime, $\tau \sim \tau + \beta$. Then, we can follow the discussion in Section 2.1 with $D = d + 1$ and $\mathcal{V} = V\beta$ and identify the exponential of the Euclidean action as $H_\tau^{\tau,1,2,\dots,d}$ of (13).

Going back to Lorentzian signature, we learn that in order to avoid the $x_*^i$ dependence, we have to choose a reference point $x_0^i$ and add a term of the Lagrangian density

$$\frac{k}{V} a_t \to \frac{k}{V} \left( a_t - \sum_i (x_*^i - x_0^i) f_{it} \right). \tag{17}$$

This added term can be viewed as a sum of $\theta$-terms in the action

$$\frac{1}{2\pi} \sum_i \theta^i \int dt\, dx^i f_{it} = \frac{1}{2\pi V} \int dt\, d^d x \sum_i \theta^i \ell^i f_{it}, \qquad \theta^i = 2\pi k \frac{x_0^i - x_*^i}{\ell^i}, \tag{18}$$

or in Euclidean space

$$\frac{1}{2\pi} \sum_i \theta^i \int d\tau\, dx^i f_{i\tau} = \frac{1}{2\pi V} \int d\tau\, d^d x \sum_i \theta^i \ell^i f_{i\tau}. \tag{19}$$

From this perspective, the choice of $x_0^i$ can be thought of as a choice of bare $\theta$-terms. However, it is crucial for us that spatial translations shift the bare $\theta$-terms. It will be important below that the periodicity of the parameter $x_0^i$ is $\frac{\ell^i}{k}$ rather than merely $\ell^i$.

As we emphasized in the Introduction, such added terms to the classical Lagrangian should not affect the classical equations of motion. Indeed, as always with $\theta$-terms, they do not contribute the equations of motion.

To summarize, the explicit $x_*^i$ dependence in the $\theta$-terms, cancels the implicit $x_*^i$ dependence in the first term. We can think of the dependence on $x_*^i$ as violation of the gauge symmetry and the added terms make the theory gauge invariant. The price we pay for that is that the dependence on $x_0^i$ explicitly breaks the translation symmetry of the problem.

### 2.2.2 Description in terms of a background gauge field

It is often useful to view every coupling constant in the theory as a background field. Here we do it for $\mathcal{L}_a$.

The $U(1)$ gauge theory has a magnetic $d-2$-form global symmetry [20] with currents and charges

$$\mathcal{J}_{\mu\nu} = \frac{1}{2\pi} f_{\mu\nu}, \qquad \partial_\mu \mathcal{J}_{\nu\rho} + \partial_\nu \mathcal{J}_{\rho\mu} + \partial_\rho \mathcal{J}_{\mu\nu} = 0, \qquad \mathcal{Q}_{\mu\nu} = \int dx^\mu dx^\nu \mathcal{J}_{\mu\nu} \in \mathbb{Z}. \tag{20}$$

Its coupling to a background $d-1$-form gauge field $A$ is through a Chern-Simons coupling

$$CS(a,A) = \frac{1}{2\pi} \int a dA \tag{21}$$

(for $d = 1$, the magnetic symmetry can be thought of as a "$-1$-form symmetry" and $A$ is a compact background scalar [20, 31, 32]).

We are interested in the theory with a constant, properly-normalized, background "magnetic" field

$$dA = \frac{2\pi k}{V} dx^1 \wedge dx^2 \cdots \wedge dx^d. \tag{22}$$

Using this background in (21), we find $\frac{k}{V} a_t$.

As is well known, Chern-Simons terms like (21) need to be defined carefully.[8] But even without doing it, we can quickly derive the breaking of continuous translation symmetry.

We are interested in a specific $A$, such that its field strength is (22). We denote the components of the $d-1$-form gauge field using its dual $A^i$, such that the gauge invariant field strength is

$$\sum_i \partial_i A^i = \frac{2\pi k}{V}. \tag{23}$$

Next, we choose a local trivialization and a gauge for $A^i$, e.g.,

$$A^1 = \frac{2\pi k}{V}(x^1 - x_0^1), \qquad A^i = 0, \quad \text{for} \quad i \neq 1, \qquad 0 \leq x^1 < \ell^1. \tag{24}$$

In this gauge, we have a discontinuity only at $x^1 = 0$. It is associated with moving from patch to patch with a transition function for $A$. That transition function is independent of $x^1$, such that we can easily explore the translation symmetry of $x^1$. (As we will see, this expression corresponds to choosing $x_0^j = 0$ for $j \neq i$.)

Regardless of the precise presentation of the Chern-Simons term, its variation under $\delta A$ is simple

$$\delta CS(a,A) = \frac{1}{2\pi} \int da \delta A. \tag{25}$$

---

[8]One way to do it, which we follow in this note, involves adding "correction terms" associated with the choice of transition functions. This is standard in general gauge theories and in particular in Chern-Simons theories. See e.g., the physics discussion in [31, 33–35], and references therein for the mathematics literature.

For $A$ of (24), a shift $x^1 \to x^1 + \epsilon^1$ leads to

$$\delta A^1 = \frac{2\pi k}{V}\epsilon^1 \,, \tag{26}$$

and hence

$$\delta CS(a,A) = \frac{k}{V}\epsilon^1 \int d^d x f_{1t} \,. \tag{27}$$

A similar conclusion appears in the other directions. (It is easier to demonstrate it using other gauges.)

It is known that despite appearance, the Chern-Simons term (21) depends on the constant mode of $A$. In our case, with a background $A$, we parameterize this mode by the reference point $x_0^i$. We see that to make our theory based on $\int d^d x a_t$ meaningful, we need to specify $d$ numbers, either the constant mode of $A$, or $x_0^i$. This leads to explicit breaking of the continuous translation symmetry.

Let us relate this discussion to the discussion in Section 2.2.1. Unlike (24), we choose a more symmetric gauge

$$A^i = \frac{2\pi k}{dV}(x^i - x_0^i), \qquad 0 \le x^i < \ell^i \,. \tag{28}$$

(We use $d$ both for the number of spatial dimensions and the exterior derivative. We hope this will not cause confusion.) Using that, the corrected $\mathcal{L}_a$ (17) is

$$\mathcal{L}_a = \frac{1}{2\pi}\sum_i \left( \partial_i A^i a_t - dA^i(x_*^i) f_{it} \right). \tag{29}$$

Here, the field strength of the background $d-1$-form gauge field is $\sum_i \partial_i A^i = \frac{2\pi k}{V}$ without a delta function at $x^i = 0$.

Using integration by parts and being careful about the surface terms, (29) can be replaced by

$$\begin{aligned}
\mathcal{L}_a &= \frac{1}{2\pi}\sum_i \left( \partial_i A^i a_t - dA^i(x_*^i) f_{it} \right) \\
&\to \frac{1}{2\pi}\sum_i \left( -A^i f_{it} - (d-1)A^i(x_*^i) f_{it} + \frac{2\pi k \ell^i}{dV} a_t \delta(x^i) \right) \\
&\to \mathcal{L}_a' = \sum_i \left( -\frac{k}{dV}(x^i - x_0^i) f_{it} + \frac{k}{dV}\ell^i \delta(x^i)\left( a_t - \left( \sum_{j\neq i}(x_*^j - x_0^j) f_{jt} \right) \right) \right).
\end{aligned} \tag{30}$$

In the first step, we used the fact that $a_t$ has a transition function only at $x_*^i$ and we dropped a total time derivative. In the second step we used the fact that the integral $\int dt\, dx^i f_{it}$ is independent of $x^j$ with $j \neq i$.

Comments about (30):

- Unlike our starting Lagrangian density $\mathcal{L}_a$ (29), away from $x^i = 0$, the new Lagrangian density $\mathcal{L}_a'$ (30) is locally gauge invariant. However, it depends explicitly on the coordinates $x^i$.

- As a consistency check, the equation of motion of $a_t$ sets the charge density to $\frac{k}{V}$ at every point, including $x^i = 0$. This is the same as in (29).

- As another consistency check, consider the $x_*^i$ dependence. $\mathcal{L}_a$ of (29) was designed such the theory is independent of $x_*^i$ – the implicit dependence in the first term is cancelled

by the explicit dependence in the second term. The same is true for $\mathcal{L}'_a$ of (30). The first term is gauge invariant and is independent of $x^i_*$. The second term is a sum of terms of the form (12) associated with co-dimension one tori labeled by $i$, such that it is also independent of $x^j_*$. (The prefactor $\frac{1}{d}$ relative to (12) is compatible with the gauge symmetry because of the contribution from the $d$ tori.)

- The violation of translation symmetry in (30) is consistent with (27).

## 2.3 Losing the translation symmetry

### 2.3.1 Breaking the translation symmetry

Let us discuss the violation of the translation symmetry in more detail.

We have already seen that the translation transformation

$$x^i \to x^i + \epsilon^i \,, \tag{31}$$

leads to

$$\mathcal{L} \to \mathcal{L} + \epsilon^i \frac{k}{V} f_{it} \,, \tag{32}$$

i.e., the translation symmetry is explicitly broken.

We interpret this explicit breaking to mean that the translation symmetry suffers from an anomaly proportional to $f_{it}$. Below, we will discuss the currents of the translation symmetry and will provide more evidence for this interpretation.

Going to Euclidean space we see that the violation of the translation symmetry is due to instantons, i.e., configurations with nonzero

$$\mathcal{Q}_{i\tau} = \frac{1}{2\pi} \int d\tau dx^i f_{i\tau} \in \mathbb{Z} \,. \tag{33}$$

They contribute to the action $2\pi k \sum_i \frac{x_0^i - x_*^i}{\ell^i} \mathcal{Q}_{i\tau}$ and hence they break the continuous translation symmetry to

$$x^i \to x^i + \frac{\ell^i}{k} \,, \tag{34}$$

i.e., the $U(1)$ translation symmetry in each direction is broken as

$$U(1) \to \mathbb{Z}_k \,. \tag{35}$$

We will soon see that the full symmetry group is not simply a product of these discrete factors.

Again, this is similar to how the Adler-Bell-Jackiw anomaly is activated by instantons and leads to an explicit breaking of the symmetry. Also, it is similar to how D-brane instantons lead to the K-theory classification of D-brane charges [36]. As in these cases, it is important that the symmetry breaking (35) is explicit symmetry breaking rather than spontaneous breaking. It does not lead to a massless Goldstone boson.

### 2.3.2 The symmetry operators

Here we discuss the symmetry operators that implement these translation symmetries. See related discussions for $d = 1$ in [18] and for $d = 2$ in [19].

First, we ignore the subtleties associated with gauge invariance. The naive momentum current, which follows from (8) is not gauge invariant[9]

$$\Theta_{tj} = \frac{k}{V}a_j + \Theta_{tj}^{(0)}, \qquad \Theta_{ij} = \delta_{ij}\frac{k}{V}a_t + \Theta_{ij}^{(0)}. \tag{37}$$

The operators $\Theta_{tj}^{(0)}$ and $\Theta_{ij}^{(0)}$ depend on $\mathcal{L}^{(0)}$ and are locally gauge invariant and translation invariant.

The momentum operator associated with the current (37)

$$p_j = \int d^d x \Theta_{tj} = \int d^d x \left(\frac{k}{V}a_j + \Theta_{tj}^{(0)}\right), \tag{38}$$

is conserved. However, it is not gauge invariant.

We can try to consider the discrete translation operators

$$\exp\left(\frac{i\ell_j}{k}p_j\right) = \exp\left(i\frac{\ell_j}{V}\int d^d x \left(a_j + \frac{V}{k}\Theta_{tj}^{(0)}\right)\right). \tag{39}$$

Again, this expression has to be defined carefully. To do that, we follow the discussion in Section 2.1 and use (13) with $D = d$ to write the generators

$$T^j = \exp\left(\frac{i\ell^j}{V}\int d^d x \left(a_j - \lambda^{(j)}\delta(x^j - x_*^j) + \sum_i (x_*^i - x_0^i)f_{ji} + \frac{V}{k}\Theta_{tj}^{(0)}\right)\right), \quad (T^j)^k = 1. \tag{40}$$

The explicit dependence of the translation generators (40) on $x_0^i$ means that they do not commute

$$T^i T^j = T^j T^i e^{\frac{2\pi i}{k}\mathcal{Q}_{ij}}, \qquad \mathcal{Q}_{ij} = \frac{1}{2\pi}\int dx^i dx^j f_{ij} = \frac{\ell^i \ell^j}{2\pi V}\int d^d x f_{ij}. \tag{41}$$

This lack of commutativity of the translation symmetry is an extension of the discrete translation group $\mathbb{Z}_k^d$ by the $d-2$-form magnetic symmetry (20). Using our nonrelativistic notation, its currents and charges are

$$\mathcal{J}_{jt} = \frac{1}{2\pi}f_{jt}, \qquad \mathcal{J}_{ij} = \frac{1}{2\pi}f_{ij},$$
$$\partial_t \mathcal{J}_{ij} = \partial_j \mathcal{J}_{it} - \partial_i \mathcal{J}_{jt}, \qquad \partial_m \mathcal{J}_{ij} + \partial_j \mathcal{J}_{mi} + \partial_i \mathcal{J}_{jm} = 0, \tag{42}$$
$$\mathcal{Q}_{ij} = \int dx^i dx^j \mathcal{J}_{ij} \in \mathbb{Z}.$$

More precisely, the translation symmetry is extended by a discrete $\mathbb{Z}_k \subset U(1)$ of the magnetic $d-2$-form symmetry. Note that this extension is not central and related to that, it does not reflect an anomaly.

Physically, the extension (41) has a simple interpretation. As we said above, the term $\frac{k}{V}a_t$ in the Lagrangian density means that all the states in the Hilbert space carry $U(1)$ charge $k$.

---

[9] A quick way to see that the gauge noninvariant part of the momentum current has this form for any $\mathcal{L}^{(0)}$ is the following. The term with $\delta_{ij}\mathcal{L}$ in $\Theta_{ij}$ of (8) leads to $\delta_{ij}\frac{k}{V}a_t$. Then, the conservation equation $\partial_t \Theta_{tj} - \sum_i \partial^i \Theta_{ij} = 0$ means that $\Theta_{tj}$ should include $\frac{k}{V}a_j$, such that the other terms, which are gauge invariant, can cancel $\frac{k}{V}f_{jt}$

$$\partial_t \Theta_{tj}^{(0)} - \sum_i \partial^i \Theta_{ij}^{(0)} = \frac{k}{V}f_{jt}. \tag{36}$$

Therefore, the lack of commutativity in (41) is the standard lack of commutativity of translations of charged particles in the presence of a magnetic field $f_{ij}$.

Unlike other situations with background magnetic field, our magnetic field is dynamical. Therefore, its flux $\mathcal{Q}_{ij}$ is an operator rather than a $c$-number. As a result, the extension in (41) is not central. Also, when the $U(1)$ gauge field is a classical background, the Hilbert space has states with various $U(1)$ charges and the lack of commutativity of the translation operators depends on the charge of the state. In our case, all the states have the same charge $k$ and therefore we have a uniform expression (41). See also a related discussion in Section 6.2.

Below, we will be interested in various commutative subgroups of our translation symmetry. Specifically, for every set of integers $k_i$ such that $k = \prod_i k_i$, the subgroup

$$\otimes_i \mathbb{Z}_{k_i} \tag{43}$$

of the translation symmetry generated by $(T^i)^{\frac{k}{k_i}}$ is not extended by the $d-2$-form symmetry.

### 2.3.3 Gauge invariant currents

As in the discussion in Section 2.2.2, we can try to replace the momentum current (37) and the momentum operator (38) by other operators that are manifestly gauge invariant, but are position dependent and perhaps even discontinuous in space. An example in $d \geq 2$ is[10]

$$\Theta'_{tj} = \frac{k}{V(d-1)} \sum_m g_m(x^m) f_{jm} + \Theta^{(0)}_{tj},$$

$$\Theta'_{ij} = \frac{k}{V(d-1)} \left( g_i(x^i) f_{jt} - \delta_{ij} \sum_m g_m(x^m) f_{mt} \right) + \Theta^{(0)}_{ij}, \tag{46}$$

$$g_m(x^m) = x^m - x_0^m, \quad \text{for} \quad 0 \leq x^m < \ell^m.$$

Compare with (28). Clearly, by shifting $x^i$ we can remove the explicit $x_0^i$ dependence and have the discontinuity at $x_0^i$.

The main point about the current (46) is that unlike the current (37), it is a well-defined, gauge invariant operator.

As we pointed out in footnote 10, the current (46) is not related to (37) by a valid improvement transformation. Still we can explore it. First, we note that, as in Section 2.2.2, integration by parts in space leads to the non-gauge invariant terms in the momentum current (37). Second, because of the discontinuities in $g_m(x^m)$, it is not conserved. Using (36), we find

$$\partial_t \Theta'_{tj} = \sum_i \partial^i \Theta'_{ij} + \frac{k}{V(d-1)} \sum_{i \neq j} \ell^i \delta(x^i) f_{jt}. \tag{47}$$

---

[10]In standard theories, with well-defined energy-momentum tensor, there is freedom to perform an "improvement transformation." Using our non-relativistic notation, it is a redefinition

$$\Theta'_{tj} = \Theta_{tj} + \sum_m \partial^m U_{mj}, \qquad \Theta'_{ij} = \Theta_{ij} + \partial_t U_{ij} + \sum_m \partial^m V_{mij}, \qquad V_{mij} = -V_{imj}. \tag{44}$$

This transformation does not change the fact that the current is conserved, i.e., $\partial_t \Theta'_{tj} = \sum_i \partial^i \Theta'_{ij}$. And with suitable boundary conditions, the total derivative $\sum_m \partial^m U_{mj}$ in $\Theta'_{tj}$ integrates to zero, such that the total momentum is unchanged.

In our case, $\Theta_{tj}$ and $\Theta_{ij}$ of (37) are not good operators. We can try to "improve" them with $U_{mj}$ and $V_{mij}$, which are also not good operators, such that the improved current $(\Theta'_{tj}, \Theta'_{ij})$ is better behaved. Specifically, ignoring the discontinuities, the expressions (46) are obtained with

$$U_{mj} = \frac{k}{V(d-1)} \left( \delta_{mj} \left( \sum_n g_n(x^n) a_n \right) - g_m(x^m) a_j \right), \qquad V_{mij} = \frac{k}{V(d-1)} \left( \delta_{jm} g_i(x^i) - \delta_{ij} g_m(x^m) \right) a_t,$$

$$g_m(x^m) = x^m - x_0^m, \quad \text{for} \quad 0 \leq x^m < \ell^m. \tag{45}$$

The left-hand-side of this equation is finite (no delta function). The explicit delta function in the right-hand-side cancels a delta function singularities in $\partial^i \Theta'_{ij}$. (Alternatively, we can delete the points $x^i = x^i_0$ and have no delta function. Then, when we integrate $\partial^i \Theta'_{ij}$ over the space, we should take into account a nonzero surface term.)

We conclude that the new current (46) is well-defined. However, it is not conserved. Similarly, unlike the momentum (38), the new momentum

$$p'_j = \int d^d x \Theta'_{tj} , \tag{48}$$

is well-defined, but it is not conserved

$$\partial_t p'_j = \frac{k}{V(d-1)} \int d^d x \sum_{i \neq j} \ell^i \delta(x^i) f_{jt} . \tag{49}$$

In order to quantify the lack of momentum conservation, we go to Euclidean time $\tau$ and find that instantons associated with nonzero $\int d\tau dx^j f_{j\tau} \in 2\pi\mathbb{Z}$ violate $p'_j$. This change in the momentum is quantized

$$\Delta p'_j = \int d\tau \partial_\tau p'_j = \frac{k}{V(d-1)} \int d\tau d^d x \sum_{i \neq j} \ell^i \delta(x_i) f_{j\tau} \in \frac{2\pi k}{\ell^j} \mathbb{Z} . \tag{50}$$

Hence, $p'_j$ is conserved only modulo $\frac{2\pi k}{\ell^j}$. Therefore, only the discrete $\mathbb{Z}_k$ translations (40) are true symmetries of the problem.

We conclude that the current $(\Theta_{tj}, \Theta_{ij})$ of (37) is conserved, but it is ill-defined, and the current $(\Theta'_{tj}, \Theta'_{ij})$ of (46) is well-defined, but not conserved. This is in accord with our interpretation of this phenomenon as an anomaly. As in the standard ABJ situation, there are two currents. One of them is conserved, but not gauge invariant, and the other is gauge invariant, but not conserved.

For all $d$, including $d = 1$, with nonzero $\Theta^{(0)}_{tj}$ we can find a similar phenomenon by replacing (46) with

$$\Theta'_{tj} = \Theta^{(0)}_{tj} , \qquad \Theta'_{ij} = \Theta^{(0)}_{ij} , \qquad \partial_t \Theta^{(0)}_{tj} - \sum_i \partial^i \Theta^{(0)}_{ij} = \frac{k}{V} f_{jt} , \tag{51}$$

where we used (36). This current is gauge invariant, but it is not conserved [18]. However, in the models discussed in Section 3, it is common to study the Lagrangian (71), where $\Theta^{(0)}_{tj} = 0$ and then the corresponding momentum operator is trivial.

## 3 Theories based on a local phase space

### 3.1 Review of some properties of phase space

#### 3.1.1 General discussion

Here we review some well known properties of symplectic manifolds and phase space. See, e.g., [37, 38] for mathematical review and [39] for a description accessible to physicists.

We consider a compact phase space $\mathcal{P}$ with local coordinates $\phi^r$. It is characterized by a closed two-form $\mathcal{F}$ (which is usually denoted by $\omega$) with quantized periods

$$\int_{\mathcal{C}} \mathcal{F} \in 2\pi\mathbb{Z} , \tag{52}$$

with $\mathcal{C}$ a closed two-cycle. Locally, $\mathcal{F} = d\mathcal{A}$. $\mathcal{F}$ is known as the symplectic structure and $\mathcal{A}$ is known as the Liouville one-form. (It is also known as the tautological one-form, the Poincare one-form, the canonical one-form, or the symplectic potential.) In terms of our coordinate system,

$$\mathcal{F} = d\mathcal{A} = \frac{1}{2}\sum_{rs}\mathcal{F}_{rs}d\phi^r \wedge d\phi^s, \qquad \mathcal{A} = \sum_r \mathcal{A}_r d\phi^r, \qquad \mathcal{F}_{rs} = \partial_r \mathcal{A}_s - \partial_s \mathcal{A}_r. \tag{53}$$

The two-form $\mathcal{F}$ is globally well-defined, but the one-form $\mathcal{A}$ is not. Using a local trivialization, as we go from patch to patch, it transforms as

$$\mathcal{A} \rightarrow \mathcal{A} + d\Lambda, \tag{54}$$

where $\Lambda$ is defined on the overlaps.

It is common to study a circle bundle $\mathcal{B}$ over $\mathcal{P}$, known as the pre-quantum bundle or the Boothby–Wang bundle [40]. The fiber is parameterized locally by $\psi \sim \psi + 2\pi$. And as we go from patch to patch, it transforms as

$$\psi \rightarrow \psi - \Lambda, \tag{55}$$

with the same $\Lambda$ as in (54). This means that the one-form

$$\alpha = \mathcal{A} + d\psi, \tag{56}$$

is globally well-defined and $d\alpha = \mathcal{F}$, i.e., $\mathcal{F}$ is exact on $\mathcal{B}$. The total space $\mathcal{B}$ is a contact manifold and the one-form $\alpha = \mathcal{A} + d\psi$ is its contact form.

Let us demonstrate this in two well-known examples.

### 3.1.2 $\mathcal{P} = \mathbb{T}^2$

$\mathbb{T}^2$ is parameterized by

$$(\phi^1, \phi^2) \sim (\phi^1 + 2\pi, \phi^2) \sim (\phi^1, \phi^2 + 2\pi). \tag{57}$$

The Liouville one-form and the symplectic structure can be taken to be

$$\mathcal{A} = \frac{1}{2\pi}\phi^1 d\phi^2, \qquad \mathcal{F} = \frac{1}{2\pi}d\phi^1 \wedge d\phi^2. \tag{58}$$

The transition functions (54) are

$$\begin{aligned}(\phi^1, \phi^2) &\rightarrow (\phi^1 + 2\pi, \phi^2), \quad \Lambda = \phi^2, \\ (\phi^1, \phi^2) &\rightarrow (\phi^1, \phi^2 + 2\pi), \quad \Lambda = 0.\end{aligned} \tag{59}$$

The circle bundle over it $\mathcal{B}$ is the Heisenberg manifold. Its coordinates are $(\phi^1, \phi^2, \psi)$ and using (55) they are subject to the identifications

$$\begin{aligned}(\phi^1 + 2\pi, \phi^2, \psi) &\sim (\phi^1, \phi^2, \psi + \phi^2), \\ (\phi^1, \phi^2 + 2\pi, \psi) &\sim (\phi^1, \phi^2, \psi), \\ (\phi^1, \phi^2, \psi + 2\pi) &\sim (\phi^1, \phi^2, \psi).\end{aligned} \tag{60}$$

Finally, the contact form (56) is

$$\alpha = \mathcal{A} + d\psi = \frac{1}{2\pi}\phi^1 d\phi^2 + d\psi. \tag{61}$$

### 3.1.3 $\mathcal{P} = S^2$

The sphere can be parameterized by a three-vector $n^A$ constrained to satisfy $\sum_A (n^A)^2 = 1$. Two convenient parameterizations are in terms of stereographic coordinates or spherical coordinates

$$
\begin{aligned}
n^1 &= \frac{z + \bar{z}}{1 + |z|^2} = \sin\vartheta \cos\varphi\,, \\
n^2 &= \frac{i(\bar{z} - z)}{1 + |z|^2} = \sin\vartheta \sin\varphi\,, \\
n^3 &= \frac{1 - |z|^2}{1 + |z|^2} = \cos\vartheta\,.
\end{aligned}
\tag{62}
$$

The standard Liouville one-form and the symplectic structure are

$$
\begin{aligned}
\mathcal{A} &= \frac{i(\bar{z}dz - zd\bar{z})}{2(1 + |z|^2)} = \frac{1}{2}(\cos\vartheta - 1)d\varphi\,, \\
\mathcal{F} &= d\mathcal{A} = -\frac{idz \wedge d\bar{z}}{(1 + |z|^2)^2} = \frac{1}{2}\sin\vartheta d\varphi \wedge d\vartheta\,.
\end{aligned}
\tag{63}
$$

The $SO(3)$ isometry of $\mathcal{P}$ acts as

$$
z \to \frac{az + b}{-\bar{b}z + \bar{a}}\,, \qquad |a|^2 + |b|^2 = 1\,, \quad (a, b) \sim (-a, -b)\,.
\tag{64}
$$

It transforms the Liouville one-form as

$$
\mathcal{A} \to \mathcal{A} + d\Lambda\,, \qquad \Lambda = \frac{i}{2}\log\left(\frac{a - b\bar{z}}{\bar{a} - \bar{b}z}\right)\,,
\tag{65}
$$

and $\mathcal{F}$ is invariant.

In this case, the pre-quantum line bundle $\mathcal{B}$ is a three-sphere $S^3$. It can be parameterized by two complex numbers $\mathcal{Z} = \begin{pmatrix} Z^1 \\ Z^2 \end{pmatrix}$ constrained to satisfy $\mathcal{Z}^\dagger \mathcal{Z} = 1$. We express them as

$$
\mathcal{Z} = \begin{pmatrix} \frac{e^{-i\psi}}{\sqrt{1 + |z|^2}} \\ \frac{e^{-i\psi}z}{\sqrt{1 + |z|^2}} \end{pmatrix}\,,
\tag{66}
$$

and identify $z$ as the coordinate above by projecting to the base (62)

$$
n^A = \mathcal{Z}^\dagger \sigma^A \mathcal{Z}\,,
\tag{67}
$$

where $\sigma^A$ are the Pauli matrices. The contact form (56) is

$$
\alpha = \mathcal{A} + d\psi = i\mathcal{Z}^\dagger d\mathcal{Z} = -id\mathcal{Z}^\dagger \mathcal{Z}\,.
\tag{68}
$$

The $SO(3)$ transformation (64), combined with $\psi \to \psi - \Lambda$ with $\Lambda$ of (65) acts as

$$
\mathcal{Z} = \begin{pmatrix} \frac{e^{-i\psi}}{\sqrt{1 + |z|^2}} \\ \frac{e^{-i\psi}z}{\sqrt{1 + |z|^2}} \end{pmatrix} \to \begin{pmatrix} \bar{a} & -\bar{b} \\ b & a \end{pmatrix} \mathcal{Z}\,.
\tag{69}
$$

This is an $SU(2)$ transformation. The identification $(a, b) \sim (-a, -b)$, which makes it an $SO(3)$ transformation is present because the transformation with $a = -1$ and $b = 0$ acts only on the fiber, which is parameterized by $\psi$.

## 3.2 Defining a theory on $\mathcal{P}$

Here we study a theory whose target space is $\mathcal{P}$. By analogy with (14), we write

$$\mathcal{L}^{Classical} = \mathcal{L}_{\mathcal{A}} + \mathcal{L}^{(0)}(\phi^r, \partial_\mu \phi^r), \qquad \mathcal{L}_{\mathcal{A}} = \frac{k}{V} \sum_r \mathcal{A}_r \partial_t \phi^r, \tag{70}$$

with $\mathcal{L}^{(0)}$ a sum of globally well-defined terms, which are manifestly translation invariant. The term $\mathcal{L}_{\mathcal{A}}$ is known as a Berry term or as a Wess-Zumino term. Soon, we will define it in the quantum theory.

We will be interested in the special case where $\mathcal{L}^{(0)}$ is independent of $\partial_t \phi^r$,

$$\mathcal{L}^{Classical} = \mathcal{L}_{\mathcal{A}} - \mathcal{H}(\phi^r, \partial_i \phi^r). \tag{71}$$

Here, $\mathcal{H}$ is the Hamiltonian density. The extension to the more general case (70) is straightforward.

A convenient approach to this problem is to embed it in a larger problem based on the larger space $\mathcal{B}$ and remove the additional field $\psi$, by imposing a gauge symmetry

$$\psi \to \psi + \lambda. \tag{72}$$

This is done by adding a dynamical $U(1)$ gauge field $a_\mu$, which transforms as

$$a_\mu \to a_\mu + \partial_\mu \lambda. \tag{73}$$

We take $a_\mu$ to be globally well-defined on $\mathcal{B}$.

Now, the Lagrangian can depend also on

$$f_{\mu\nu} = \partial_\mu a_\nu - \partial_\nu a_\mu, \qquad X_\mu = \partial_\mu \psi - a_\mu + \sum_r \mathcal{A}_r \partial_\mu \phi^r. \tag{74}$$

They are gauge invariant and because of (54) and (55) they are single valued across overlaps between patches. In addition, as in Section 2, we add the term $\frac{k}{V} a_t$. In fact, as we will soon see, when we do that, we do not need to include $\mathcal{L}_{\mathcal{A}}$ in (70).

We end up with a Lagrangian on $\mathcal{B}$ with a $U(1)$ gauge field $a_\mu$ and a term $\frac{k}{V} a_t$. Therefore, we can follow the discussion in Section 2.

To relate to the original problem on $\mathcal{P}$, without $\psi$ and $a_\mu$, we arrange the Lagrangian such that at low energies,

$$X_\mu = 0. \tag{75}$$

For example, we can take (75) to be a constraint, which can be imposed by adding a real Lagrange multiplier field $Y^\mu$ with the term $\sum_\mu Y^\mu X_\mu$ in the Lagrangian density.

Now, we can set $a_\mu = \sum_r \mathcal{A}_r \partial_\mu \phi^r + \partial_\mu \psi$ (i.e., the pullback to spacetime of the contact form (56)). At this stage, $\psi$ is the only field transforming under the gauge symmetry (72), so locally, we can set it to zero. The only places it should be analyzed carefully is in the term $\mathcal{L}_a = \frac{k}{V} a_t$ and in various operators that depend explicitly on $a_\mu$.

We end up substituting

$$a_\mu \to \sum_r \mathcal{A}_r \partial_\mu \phi^r + \partial_\mu \psi, \tag{76}$$

and therefore, $f_{\mu\nu} \to \sum_{rs} \mathcal{F}_{rs} \partial_\mu \phi^r \partial_\nu \phi^s$. Locally, this leads to a Lagrangian density of the form (70) or its special case (71).

Conversely, if we are interested in the theory on $\mathcal{P}$, we start with a Lagrangian density and operators, which can include $\sum_r \mathcal{A}_r \partial_\mu \phi^r$. To define them properly, we add the field $\psi$ and substitute

$$\sum_r \mathcal{A}_r \partial_\mu \phi^r \to \sum_r \mathcal{A}_r \partial_\mu \phi^r + \partial_\mu \psi, \tag{77}$$

which affects it only globally. In other words, we add the field $\psi$ and replace the pull-back of $\mathcal{A}$ by the pull-back of the contact form $\alpha$, which is globally well-defined. Then, we remove $\psi$ by imposing the $U(1)$ gauge symmetry. We do that by noting that the pull-back of $\alpha$ is a $U(1)$ gauge field $a_\mu$ and then we can follow the discussion in Section 2 with all the added terms that it leads to.

In the following sections, we will demonstrate this procedure explicitly. First, we will do it in quantum mechanics, i.e., $d = 0$ and then in field theory. We will use the examples of $\mathbb{T}^2$ in Section 3.1.2 and $S^2$ in Section 3.1.3, because they appear in the condensed matter applications and they exhibit special new elements.

## 3.3 Review of quantum mechanics on $\mathcal{P}$

### 3.3.1 General discussion

Here we analyze the Lagrangian in Section 3.2 in the special case of quantum mechanics, i.e. $d = 0$. We write it as

$$\mathcal{L}^{Classical} = k \sum_r \mathcal{A}_r(\phi^s)\partial_t \phi^r, \qquad k \in \mathbb{Z}. \tag{78}$$

For simplicity, we set the Hamiltonian $\mathcal{H}$ to zero.

We would like to find the quantum theory based on the classical Lagrangian (78). We can always do it along the lines of Section 3.2. But in the special case where the phase space $\mathcal{P}$ is simply connected, there is another definition [23]. We take time to be a Euclidean circle $\tau \sim \tau + \beta$ and extend it to a disk with the other direction parameterized by $u$. Then, we extend the dynamical variables $\phi^r$ into $\mathcal{D}$ and define the Euclidean action as

$$\mathcal{S}_{Euclidean} = ik \int_{\mathcal{D}} d\tau du \sum_{rs} \mathcal{F}_{rs}\partial_u \phi^r \partial_\tau \phi^s. \tag{79}$$

$k$ has to be an integer for the integrand in the functional integral $e^{-\mathcal{S}_{Euclidean}}$ to be independent of the extension of $\phi^r$ into $\mathcal{D}$.

This definition cannot be used when $\mathcal{P}$ is not simply connected.[11] In that case, configurations in Euclidean time that wind around non-contractible cycles in $\mathcal{P}$ cannot be extended into $\mathcal{D}$. Such configurations can be interpreted as instantons, and related to that, the theory depends on $\theta$-parameters associated with that winding. In this case, we cannot use (79) and we have to use the procedure in Section 3.2. We will demonstrate it in Section 3.3.2. In Section 3.4, we will see that in higher dimensions, a definition like (79) is never possible.

In preparation for the later sections, we write the commutation relations

$$[\phi^r, \phi^s] = \frac{i}{k}\mathcal{F}^{rs}, \qquad \sum_s \mathcal{F}^{rs}\mathcal{F}_{su} = \delta^r_u. \tag{80}$$

Note that the phase space coordinates $\phi^r$ are not good operators because they are not globally well-defined in the phase space. Still we can use these commutation relations when we manipulate well-defined operators.

In the next two sections, we will demonstrate this discussion with two well-known examples, $\mathcal{P} = \mathbb{T}^2$ and $\mathcal{P} = S^2$.

### 3.3.2 Fuzzy $\mathbb{T}^2$

Using the results in Section 3.1.2, the Lagrangian (78) is

$$\mathcal{L}^{Classical}_{\mathbb{T}^2} = \frac{k}{2\pi}\phi^1 \partial_t \phi^2. \tag{81}$$

---

[11]We thank Edward Witten for an interesting discussion about non-simply-connected $\mathcal{P}$.

Classically, the system has a global $U(1)^{(1)} \times U(1)^{(2)}$ symmetry acting as $\phi^r \to \phi^r + \epsilon^r$ and a $\mathbb{Z}_4$ duality symmetry generated by

$$(\phi^1, \phi^2) \to (\phi^2, -\phi^1). \tag{82}$$

We will soon see what happens to these symmetries in the quantum theory.

The Lagrangian (81) is known to describe a particle moving on a two-dimensional torus parameterized by $\phi^r \sim \phi^r + 2\pi$ in the lowest Landua level of a constant magnetic field $B = \frac{k}{2\pi}$. In this context, the $U(1)^{(1)} \times U(1)^{(2)}$ classical symmetry is the translation symmetry of the spatial torus and the $\mathbb{Z}_4$ duality symmetry (82) is spatial rotation.

Let us start with canonical quantization. We have the commutation relations (80)

$$[\phi^1, \phi^2] = -\frac{2\pi i}{k}. \tag{83}$$

The operators in the quantum theory correspond to well-defined functions on the phase space. They are generated by

$$U_1 = e^{i\phi^1}, \qquad U_2 = e^{i\phi^2}, \tag{84}$$

and satisfy

$$U_1 U_2 = e^{\frac{2\pi i}{k}} U_2 U_1. \tag{85}$$

Hence, the operators $U_1^k$ and $U_2^k$ commute with all the other operators and we can take them to be the unit operators.[12] As a result, the Hilbert space has $k$ states and the operators $U_1$ and $U_2$ can be represented as clock and shift operators

$$(U_1)_{IJ} = \delta_{I,J} e^{\frac{2\pi i (I-1)}{k}}, \qquad (U_2)_{IJ} = \delta_{I, J+1 \bmod k}, \qquad I, J = 1, \ldots, k. \tag{86}$$

We conclude that $U_1$ and $U_2$ generate a global $\mathbb{Z}_k^{(1)} \times \mathbb{Z}_k^{(2)}$ symmetry. Furthermore, this global symmetry is realized projectively. Adding the duality symmetry (82), we have $\mathbb{Z}_k^{(1)} \times \mathbb{Z}_k^{(2)} \rtimes \mathbb{Z}_4$. In the representation (86), it is generated by the generalized Walsh–Hadamard matrix

$$W_{IJ} = \frac{1}{\sqrt{k}} e^{\frac{2\pi i (I-1)(1-J)}{k}}, \qquad W U_1 W^{-1} = U_2, \qquad W U_2 W^{-1} = U_1^{-1}. \tag{87}$$

Let us turn now to the Euclidean path integral description of this theory. In this case, $\mathcal{P}$ is not simply connected and we cannot use the definition (79). We have to use the discussion in Section (3.2). To make it more concrete, we will use a choice of local trivialization. (For background material for this discussion, see e.g., [31, 33–35] for presentations for physicists, and references therein for the mathematics literature.) We cover the Euclidean time circle $\tau \sim \tau + \beta$ with patches with transition functions between them. In each patch, $\phi^r$ are real numbers and the transition functions are in $2\pi\mathbb{Z}$.

For example, we can pick a point $\tau_*$ and view $\phi^r$ as maps from the circle minus the point $\tau_*$ to real numbers.[13] At $\tau_*$, we have transition functions $\phi^r(\tau_*^+) = \phi^r(\tau_*^-) - 2\pi \mathcal{W}_\tau^r$ with $\mathcal{W}_\tau^r \in \mathbb{Z}$. This corresponds to

$$\int d\tau \, \partial_\tau \phi^r = 2\pi \mathcal{W}_\tau^r. \tag{88}$$

Clearly, $\mathcal{W}_\tau^r$ are the winding numbers around the Euclidean time $\tau$ and the configurations with nonzero $\mathcal{W}_\tau^r$ are instantons.

---

[12]More generally, they can be taken to be c-number phases. This can be represented by writing in the Lagrangian $\frac{1}{2\pi}(\theta^1 \partial_t \phi^1 + \theta^2 \partial_t \phi^2)$. As we will soon see, the appearance of these $\theta$-parameters is related to the explicit breaking of $U(1)^{(1)} \times U(1)^{(2)}$.

[13]More precisely, as in footnote 6, we extend the patches beyond $\tau_*$ and use the transition function in the overlap region.

Now we can follow the presentation in Section (3.2). Since the fields $\phi^r$ jump at $\tau_*$ by $2\pi\mathcal{W}^r_\tau$, equation (54) tells us that $\Lambda = \mathcal{W}^1_\tau\phi^2$ and hence, $\psi$ jumps by $-\mathcal{W}^1_\tau\phi^2$. Then, the substitution (77) leads to

$$
\begin{aligned}
\mathcal{S}_{Euclidean\ \mathbb{T}^2} &= -\frac{ik}{2\pi}\int d\tau\, a_\tau = -\frac{ik}{2\pi}\int d\tau\left(\phi^1\partial_\tau\phi^2 + \partial_\tau\psi\right)\\
&= -\frac{ik}{2\pi}\int d\tau\left(\phi^1\partial_\tau\phi^2 - 2\pi\mathcal{W}^1_\tau\phi^2\delta(\tau-\tau_*)\right).
\end{aligned}
\tag{89}
$$

Comments:

- The added term with $\phi^2(\tau_*)$ can be written either as $\phi^2(\tau_*^+)$ or as $\phi^2(\tau_*^-)$. The difference between them does not affect $e^{-\mathcal{S}_{Euclidean}}$.

- It is easy to check that $e^{-\mathcal{S}_{Euclidean}}$ is independent of the trivialization. In particular, it is independent of the point $\tau_*$.

- In writing (89), we chose a trivialization of the $U(1)$ gauge field without a transition function. Therefore, the term with $\lambda^{(\tau)}$ in (9) is not needed.

- The fact that the dependence on the added field $\psi$ drops out of (89) is in accord with its gauge symmetry.

- $e^{-\mathcal{S}_{Euclidean}}$ is invariant under the $\mathbb{Z}_4$ duality symmetry (82).

- $e^{-\mathcal{S}_{Euclidean}}$ is not invariant under the classical $U(1)^{(1)} \times U(1)^{(2)}$ symmetry acting as $\phi^r \to \phi^r + \epsilon^r$. Such a transformation shifts the Euclidean action by $-ik(\epsilon^1\mathcal{W}^2_\tau - \epsilon^2\mathcal{W}^1_\tau)$. This is the same as shifting the $\theta$-parameters in footnote 12. (Note the similarity to our discussion of the breaking of translation symmetry in (32).) We see that instantons of $\phi^1$ carry charge $+k\mathcal{W}^1_\tau$ under $U(1)^{(2)}$ and instantons of $\phi^2$ carry charge $-k\mathcal{W}^2_\tau$ under $U(1)^{(1)}$. As a result, only $\mathbb{Z}_k^{(1)} \times \mathbb{Z}_k^{(2)} \subset U(1)^{(1)} \times U(1)^{(2)}$ is a global symmetry.

- This breaking of $U(1)^{(1)} \times U(1)^{(2)}$ by instantons can also be seen as follows. The equations of motion following from the Euclidean action (89) state that $\partial_\tau\phi^r = 0$ leading to vanishing instanton number $\mathcal{W}^r_\tau = 0$. More generally, with insertion of operators, e.g., $e^{i\phi^r}$, the equations of motion lead to discontinuities in $\phi^r$ such that the total winding numbers $\mathcal{W}^r_\tau$ is correlated with the violation of $U(1)^{(1)} \times U(1)^{(2)}$.

- We interpret this breaking of $U(1)^{(1)} \times U(1)^{(2)} \to \mathbb{Z}_k^{(1)} \times \mathbb{Z}_k^{(2)}$ as an ABJ anomaly associated with instantons.

- Another aspect of the instantons is that the sum over them, i.e., the sum over $\mathcal{W}^r_\tau$, constrains $\phi^r = \frac{2\pi}{k}\mathbb{Z}$, reflecting the $\mathbb{Z}_k^{(1)} \times \mathbb{Z}_k^{(2)}$ global symmetry.

In conclusion, the fact that $\mathcal{P}$ is not simply connected leads to instantons. These instantons break the classical internal $U(1)^{(1)} \times U(1)^{(2)}$ symmetry to a discrete subgroup $\mathbb{Z}_k^{(1)} \times \mathbb{Z}_k^{(2)}$. This is similar to our discussion in Section (2.3), where instantons break the translation symmetry to a discrete subgroup. Unlike the case in Section (2.3), here the breaking $U(1)^{(1)} \times U(1)^{(2)} \to \mathbb{Z}_k^{(1)} \times \mathbb{Z}_k^{(2)}$ is of a symmetry of the target space, i.e., an internal symmetry, rather than breaking of a spatial symmetry. We will return to this point in Section 3.4.1.

### 3.3.3 Fuzzy $S^2$

This problem is familiar from the study of a particle on a sphere surrounding a magnetic monopole. Using the results in Section 3.1.3, the Lagrangian (78) is

$$\mathcal{L}_{S^2}^{Classical} = \frac{ik}{2} \frac{\bar{z}\partial_t z - z\partial_t \bar{z}}{1 + |z|^2} = \frac{k}{2}(\cos\vartheta - 1)\partial_t \varphi \,. \tag{90}$$

Let us turn to the quantum theory. Since in this case the phase space is simply connected, we can define the action using (79), without the need for the more complicated procedure. Related to that, no "correction terms" are needed.

Then, this Lagrangian describes a single $SU(2)$ representation with spin $s = \frac{k}{2}$. The global symmetry of the system is $SO(3)$ and it is realized projectively for odd $k$.

It is also worth noting that the Lagrangian (90) is not invariant under the global $SO(3)$ transformation (64). Instead,

$$\mathcal{L}_{S^2}^{Classical} = \frac{ik}{2} \frac{\bar{z}\partial_t z - z\partial_t \bar{z}}{1 + |z|^2} \rightarrow \mathcal{L}_{S^2}^{Classical} + \frac{ik}{2}\partial_t \log\left(\frac{a - b\bar{z}}{\bar{a} - \bar{b}z}\right) \,. \tag{91}$$

Demanding that the total time derivative integrates to $2\pi\mathbb{Z}$ is another way to see the quantization of $k$.

Even though it is not necessary here, we would like to comment that in this case, the procedure of Section 3.2 is quite familiar to physicists and it is known as the $CP^1$ presentation of the model. We start with $\mathcal{B} = S^3$, view it as an $S^1$ bundle over $S^2$ and then gauge the $U(1)$ that acts along the fiber.

## 3.4 Field theory on $\mathcal{P}$

### 3.4.1 General discussion

In this section, we will explore the field theory defined in Section 3.2. We will study the Lagrangian density (70), and will focus on the special case (71) where $\mathcal{L}^{(0)}$ is independent of $\partial_t \phi^r$

$$\mathcal{L}^{Classical} = \frac{k}{V}\left(\sum_r \mathcal{A}_r(\phi^s)\partial_t \phi^r\right) - \mathcal{H}(\phi^r, \partial_i \phi^r)\,, \tag{92}$$

or its Euclidean version

$$\mathcal{L}_{Euclidean}^{Classical} = \left(-\frac{ik}{V}\left(\sum_r \mathcal{A}_r(\phi^s)\partial_\tau \phi^r\right) + \mathcal{H}(\phi^r, \partial_i \phi^r)\right)\,. \tag{93}$$

As in the discussion following (77), in the quantum theory, we need to add to (92) and (93) appropriate terms to make them well-defined.

In Section 3.3, we mentioned that if the phase space $\mathcal{P}$ is not simply connected, we cannot define the action as in (79) because the fields cannot be extended to the added dimension parameterized by $u$. A similar issues arises in the field theory (93) for any phase space $\mathcal{P}$. The reason is that the phase space of the field theory $\mathcal{P}_d$ is the space of maps from our toroidal space $\mathbb{T}^d$ to $\mathcal{P}$. (Using this notation, the phase space of the quantum mechanical problem can be denoted $\mathcal{P}_0$.) This space is never simply connected.

To see that $\mathcal{P}_d$ is never simply connected, recall that $\mathcal{P}$ has a nontrivial two-cycle dual to the symplectic structure $\mathcal{F}$. Consider a one-parameter family of maps from a spatial cycle $j$ (the cycle parameterized by $x^j \sim x^j + \ell^j$) to the target space $\mathcal{P}$ that wraps the two-cycle in $\mathcal{P}$ dual to $\mathcal{F}$. Clearly, this one-parameter family of maps is not contractible. As a result, $\mathcal{P}_d$ is not simply connected and we cannot use (79).

Related to that, the Euclidean space functional integral includes instantons. These are configuration where the two-torus parameterized by $\tau \sim \tau + \beta$ and $x^j \sim x^j + \ell^j$ is mapped to the two-cycle mentioned above. The corresponding instanton number is

$$\mathcal{Q}_{i\tau} = \frac{1}{2\pi} \int dx^i d\tau \sum_{rs} \mathcal{F}_{rs} \partial_i \phi^r \partial_\tau \phi^s \,. \tag{94}$$

In the description of this theory as a $U(1)$ gauge theory in Section 3.2, these instantons are the gauge theory instantons with nonzero $\int dx^i d\tau f_{i\tau}$. From that perspective, they are the same instantons we discussed around (33) and they have here the same consequences as there.

First, they are associated with $\theta$-terms (18)

$$\sum_i \frac{\theta^i}{2\pi} \int dx^i dt \sum_{rs} \mathcal{F}_{rs} \partial_i \phi^r \partial_\tau \phi^s = \sum_i \frac{\theta^i \ell^i}{2\pi V} \int d^d x dt \sum_{rs} \mathcal{F}_{rs} \partial_i \phi^r \partial_\tau \phi^s \,. \tag{95}$$

Second, they make the definition of the first term in the Lagrangian $\frac{k}{V} \sum_r \mathcal{A}_r(\phi^s) \partial_t \phi^r$ more complicated.

Third, as in (42), for $d \geq 2$, they are related to a $d-2$-form global symmetry [20]

$$\mathcal{J}_{jt} = \frac{1}{2\pi} \sum_{rs} \mathcal{F}_{rs} \partial_j \phi^r \partial_t \phi^s \,, \qquad \mathcal{J}_{ij} = \frac{1}{2\pi} \sum_{rs} \mathcal{F}_{rs} \partial_i \phi^r \partial_j \phi^s \,,$$
$$\partial_t \mathcal{J}_{ij} = \partial_j \mathcal{J}_{it} - \partial_i \mathcal{J}_{jt} \,, \qquad \partial_m \mathcal{J}_{ij} + \partial_j \mathcal{J}_{mi} + \partial_i \mathcal{J}_{jm} = 0 \,, \tag{96}$$
$$\mathcal{Q}_{ij} = \int dx^i dx^j \mathcal{J}_{ij} \in \mathbb{Z} \,.$$

And most important, they break the $U(1)^d$ translation symmetry to $\mathbb{Z}_k^d$, which is extended using the $d-2$-form symmetry (96).

So far, we have not used the equations of motion of (92). They are

$$\frac{k}{V} \sum_r \mathcal{F}_{sr} \partial_t \phi^r = \frac{\partial \mathcal{H}}{\partial \phi^s} - \sum_i \partial_i \left( \frac{\partial \mathcal{H}}{\partial(\partial_i \phi^s)} \right) \,. \tag{97}$$

As a check, they are globally well-defined. We will now study some of their consequences.

Let us start with the $d-2$-form internal symmetry (96), which is valid without using the equations of motion. Using (97),

$$\mathcal{J}_{jt} = \frac{1}{2\pi} \sum_{rs} \mathcal{F}_{rs} \partial_j \phi^r \partial_t \phi^s = \sum_i \partial^i \widehat{\mathcal{J}}_{ij} \,, \qquad \widehat{\mathcal{J}}_{ij} = \frac{V}{2\pi k} \left( \delta_{ij} \mathcal{H} - \sum_s \frac{\partial \mathcal{H}}{\partial(\partial_i \phi^s)} \partial_j \phi^s \right) \,, \tag{98}$$

i.e., $\mathcal{J}_{jt}$ is a total derivative of a globally well-defined operator $\widehat{\mathcal{J}}_{ij}$. This conclusion follows from the assumption that $\mathcal{L}^{(0)}$ in (70) is independent of $\partial_t \phi^r$. Note that even though the $d-2$-form symmetry (96) exists only for $d \geq 2$, equation (98) is valid also for $d = 1$. In that case, it can be thought of as a "−1-form symmetry."

As a result of (98),

$$\int d^d x \mathcal{J}_{jt} = 0 \,, \tag{99}$$

and the conservation equation (96) can be written as

$$\partial_t \mathcal{J}_{ij} = \sum_k \left( \partial_j \partial^k \widehat{\mathcal{J}}_{ki} - \partial_i \partial^k \widehat{\mathcal{J}}_{kj} \right) \,. \tag{100}$$

For $d = 2$, this becomes $\partial_t \mathcal{J}_{12} = \partial_1 \partial_2 (\widehat{\mathcal{J}}_{11} - \widehat{\mathcal{J}}_{22}) + \partial_2 \partial_2 \widehat{\mathcal{J}}_{21} - \partial_1 \partial_1 \widehat{\mathcal{J}}_{12}$, which can be interpreted as the conservation of a dipole current [16].[14]

Next, we turn to the spacetime symmetries. Since we take $\mathcal{H}$ to be time independent, we have the conserved energy current (8). Using the equations of motion (97),

$$\Theta_{tt} = \mathcal{H}, \qquad \Theta_{it} = \sum_r \frac{\partial \mathcal{H}}{\partial (\partial_i \phi^r)} \partial_t \phi^r, \qquad \partial_t \Theta_{tt} = \sum_i \partial^i \Theta_{it}. \tag{101}$$

Similarly, for the translation symmetry, (8) leads to

$$\Theta_{tj} = \frac{k}{V} \sum_r \mathcal{A}_r \partial_j \phi^r,$$

$$\Theta_{ij} = \sum_r \frac{\partial \mathcal{H}}{\partial (\partial_i \phi^r)} \partial_j \phi^r + \delta_{ij} \mathcal{L}^{Classical} = \delta_{ij} \frac{k}{V} \sum_r \mathcal{A}_r \partial_t \phi^r - \frac{2\pi k}{V} \widehat{\mathcal{J}}_{ij}, \tag{102}$$

$$\partial_t \Theta_{tj} = \sum_i \partial^i \Theta_{ij},$$

where we used the notation (98). Comparing with (37) and the substitution (77), we see that $\Theta_{tj}^{(0)}$ in (37) does not contribute because we took $\mathcal{L}^{(0)}$ in (70) to be independent of $\partial_t \phi^r$. This is related to the fact (98) that because of this assumption about $\mathcal{L}^{(0)}$, $\mathcal{J}_{jt}$ is a spatial derivative of a well-defined operator $\widehat{\mathcal{J}}_{ij}$. See also [16].

Crucially, unlike the energy current (101), the momentum current (102) is not globally well-defined. Even its integrated version $\int d^d x \Theta_{tj}$, i.e., the momentum operator is not meaningful. In the context of the model on $\mathcal{P} = S^2$, which will be discussed in Section 3.4.3, this point was first noted in [6] (see also [54–56] for earlier discussion).

Now, we can repeat the discussion in Section 2.3 using the substitution (77). Instead, we will simply summarize the conclusions.

The translation operators are of the form

$$T^j = \exp\left( \frac{i\ell^j}{k} \int d^d x \, \Theta_{tj} + \dots \right) = \exp\left( \frac{i\ell^j}{V} \int d^d x \sum_r \mathcal{A}_r \partial_j \phi^r + \dots \right). \tag{103}$$

The ellipses remind us that we should add "correction terms" to define these expressions more carefully. The exponent should be defined in terms of $a_j$ as in (77) and then corrected as in Section 2. We will demonstrate it in examples below.

Then, as in Section 2.3, these translation operators have the following properties:

- $T^j$ generate a discrete $\mathbb{Z}_k$ translation in each direction

$$(T^j)^k = 1. \tag{104}$$

  Instantons, i.e., Euclidean configurations with nonzero $\mathcal{Q}_{j\tau}$ of (94), explicitly break the continuous translation symmetry to this discrete subgroup.

- This discrete translation symmetry is extended by a $\mathbb{Z}_k \subset U(1)$ $d-2$-form symmetry (96)

$$T^i T^j = T^j T^i \exp\left( \frac{2\pi i}{k} \mathcal{Q}_{ij} \right). \tag{105}$$

  A related non-commutativity of translations in the model based on $\mathcal{P} = S^2$ was noted in the infinite volume theory in [12, 16].

---

[14]Dipole symmetries have been discussed by many authors including [24, 41–53] and it was pointed in [47, 52, 53], that in compact spaces, they are discrete. This discreteness is closely related to our discrete translation symmetry.

- For every set of integers $k_i$ such that $k = \prod_i k_i$, the subgroup

$$\otimes_i \mathbb{Z}_{k_i} \subset \mathbb{Z}_k^d \,, \tag{106}$$

generated by $(T^i)^{\frac{k}{k_i}}$ is Abelian and is not extended by the $d-2$-form symmetry. This fact will be important in Section 5.

- As in Section 2.3.3, for $d \geq 2$, we can find another expression for the momentum current

$$\Theta'_{tj} = \frac{2\pi k}{V(d-1)} \sum_m g_m(x^m) \mathcal{J}_{jm}\,,$$

$$\Theta'_{ij} = \frac{2\pi k}{V(d-1)} \left( g_i(x^i) \mathcal{J}_{jt} - \delta_{ij} \sum_m g_m(x^m) \mathcal{J}_{mt} - (d-1)\widehat{\mathcal{J}}_{ij} \right), \tag{107}$$

$$g_m(x^m) = x^m - x_0^m\,, \quad \text{for} \quad 0 \leq x^m < \ell^m$$

(compare with (46)). Clearly, by shifting $x^m$ we can remove the explicit $x_0^i$ dependence and have the discontinuity at $x_0^i$. It is well-defined in the target space, but it is not continuous in space. Then, this current is not conserved at its discontinuity. The infinite volume limit of this current, which does not exhibit the discontinuity, but has bad behavior at infinity was discussed in the context of the model on $\mathcal{P} = S^2$ in [8, 13, 16]. Since it is in infinite volume, it does not expose the discrete translation symmetry generated by $T^j$.

Let us summarize these conclusions along the lines of Section 2.1. As in (9), we start by studying

$$H(\mathcal{A})_\mu = \exp\left( i \int dx^\mu \left( \sum_r \mathcal{A}(\phi)_r \partial_\mu \phi^r + \dots \right) \right). \tag{108}$$

For simply connected $\mathcal{P}$, this can be done by adding a bulk. Otherwise, some correction terms, represented by the ellipses, are needed.

Then, we would like to integrate the logarithm of $H(\mathcal{A})_\mu$ over other directions. Since this logarithm has a $2\pi\mathbb{Z}$ ambiguity, we view it as real and let it jump at $x_*^\nu$. Crucially, the results depend on $x_*^\nu$. To remove this dependence, we choose a reference point $x_0^\nu$ and as in (13), we write

$$H(\mathcal{A})_\mu^{(1,2,\dots,D)} = \exp\left( \frac{i\ell^\mu}{V} \int d^D x \left( \mathcal{A}(\phi)_r \partial_\mu \phi^r + \dots + \sum_\nu (x_*^\nu - x_0^\nu) \sum_{rs} \mathcal{F}_{rs} \partial_\mu \phi^r \partial_\nu \phi^s \right) \right). \tag{109}$$

In quantum mechanics, i.e., $d = 0$, only (108) is needed. For defining the theory for $d \geq 1$, (109) is needed and hence the breaking of the translation symmetry. For defining the discrete translation operator in $d = 1$, we can use (108). But for $d \geq 2$, we need (109), which leads to the lack of commutativity of the discrete translation operators.

### 3.4.2 Fuzzy $\mathbb{T}^2$

In this case, the Lagrangian (92) is

$$\mathcal{L}_{\mathbb{T}^2}^{Classical} = \frac{k}{2\pi V} \phi^1 \partial_t \phi^2 - \mathcal{H}(\phi^r, \partial_i \phi^r)\,, \tag{110}$$

and hence

$$[\phi^1(\vec{x}, t), \phi^2(\vec{y}, t)] = -\frac{2\pi i V}{k} \delta^{(d)}(\vec{x} - \vec{y})\,. \tag{111}$$

Classically, the symmetry of the first term is $U(1)^{(1)} \times U(1)^{(2)}$. However, as in the discussion in Section 3.3.2, the symmetry is actually smaller. Focusing on the zero modes of $\phi^1$ and $\phi^2$, the symmetry of this term is only $\mathbb{Z}_k^{(1)} \times \mathbb{Z}_k^{(2)}$. We will take the Hamiltonian such that it is invariant under $\phi^r \to \phi^r + 2\pi$, i.e., it is well defined, but it can break that $\mathbb{Z}_k^{(1)} \times \mathbb{Z}_k^{(2)}$ symmetry. We will return to this point in Section 5.

In addition to the $d-2$-form global symmetry (96), this theory also has a $U(1) \times U(1)$ $d-1$-form winding symmetries

$$\mathcal{J}_i^r = \frac{1}{2\pi}\partial_i \phi^r, \qquad \mathcal{J}_t^r = \frac{1}{2\pi}\partial_t \phi^r, \qquad \partial_j \mathcal{J}_i^r = \partial_i \mathcal{J}_j^r, \qquad \mathcal{W}_i^r = \int dx^i \mathcal{J}_i^r \in \mathbb{Z}. \tag{112}$$

In fact, these $d-1$-form symmetries, imply the $d-2$-form symmetry (96)

$$\begin{aligned}
\mathcal{J}_{it} &= \mathcal{J}_i^1 \mathcal{J}_t^2 - \mathcal{J}_t^1 \mathcal{J}_i^2, \\
\mathcal{J}_{ij} &= \mathcal{J}_i^1 \mathcal{J}_j^2 - \mathcal{J}_j^1 \mathcal{J}_i^2, \\
\mathcal{Q}_{ij} &= \mathcal{W}_i^1 \mathcal{W}_j^2 - \mathcal{W}_j^1 \mathcal{W}_i^2.
\end{aligned} \tag{113}$$

(Such relations between higher-form symmetries are quite standard. See, e.g., [57].) Also, (98) is still valid and therefore the equations of motion imply that

$$\int d^d x (\mathcal{J}_i^1 \mathcal{J}_t^2 - \mathcal{J}_t^1 \mathcal{J}_i^2) = 0. \tag{114}$$

As always, this operator equation is violated in the presence of other operator insertions.

Finally, the first term in (110) has the $\mathbb{Z}_4$ duality symmetry (82). If the Hamiltonian $\mathcal{H}$ has that symmetry, then the full theory is $\mathbb{Z}_4$ invariant, i.e., it is self-dual.

Let us define the theory more carefully. The configurations in the Euclidean functional integral fall into classes labeled by the winding numbers

$$\mathcal{W}_\mu^r = \frac{1}{2\pi}\int dx^\mu \partial_\mu \phi^r \in \mathbb{Z}, \qquad \mu = \tau, i. \tag{115}$$

To define their action, we follow our approach above and choose a local trivialization. This means that we pick a point $x_*^\mu$ and represent the configurations as

$$\phi^r = 2\pi \mathcal{W}_\tau^r \frac{\tau}{\beta} + 2\pi \sum_i \mathcal{W}_i^r \frac{x^i}{\ell^i} + \tilde{\phi}^r, \qquad x_*^i \le x^i < x_*^i + \ell^i, \quad \tau_* \le \tau < \tau_* + \beta, \tag{116}$$

with $\tilde{\phi}^r$ periodic spacetime-dependent real functions (as opposed to circle-valued). Then, we can follow the discussions around (13) and around (89) and write the Euclidean action

$$\begin{aligned}
S_{Euclidean\ \mathbb{T}^2} = \int d\tau d^d x \Bigg[ &-\frac{ik}{2\pi V}\Big(\phi^1 \partial_\tau \phi^2 - 2\pi \mathcal{W}_\tau^1 \phi^2 \delta(\tau - \tau_*) \\
&+ \sum_i (\partial_\tau \phi^1 \partial_i \phi^2 - \partial_\tau \phi^2 \partial_i \phi^1)(x_*^i - x_0^i)\Big) + \mathcal{H}\Bigg].
\end{aligned} \tag{117}$$

This corresponds to the Lorentzian Lagrangian density

$$\mathcal{L}_{\mathbb{T}^2} = \frac{k}{2\pi V}\left(\phi^1 \partial_t \phi^2 + \sum_i (\partial_t \phi^1 \partial_i \phi^2 - \partial_t \phi^2 \partial_i \phi^1)(x_*^i - x_0^i)\right) - \mathcal{H}. \tag{118}$$

Again, the added terms to $\mathcal{L}_{\mathbb{T}^2}^{Classical}$ do not affect the equations of motion.

As in Sections 2.1 and 3.3.2, the $\tau_*$ dependence in (117) is cancelled between the first and the second term. And as in Section 2.1, the $x_*^i$ dependence is cancelled by the third term. Similarly, (118) is independent of $x_*^i$. However, the final result depends on the chosen reference point $x_0^i$ and its interpretation is as in the previous sections.[15]

Once we defined the action, we can write the translation operators. Making the corrected version of (103) explicit for this case, we have

$$T^j = \exp\left[\frac{i\ell^j}{2\pi V}\int d^d x\left(\phi^1\partial_j\phi^2 - 2\pi\mathcal{W}_j^1\phi^2\delta(x^j - x_*^j)\right)\right.$$
$$\left. + 2\pi i\sum_i(\mathcal{W}_j^1\mathcal{W}_i^2 - \mathcal{W}_j^2\mathcal{W}_i^1)\frac{x_*^i - x_0^i}{\ell^i}\right]. \tag{119}$$

Again, $T^j$ is independent of $x_*^i$, but it does depend on $x_0^i$.

Interestingly, with the added terms in (117), (118), and (119), these expressions are invariant under the $\mathbb{Z}_4$ duality symmetry $(\phi^1, \phi^2) \to (-\phi^2, \phi^1)$ of (82).

Let us discuss the dependence on $x_0^i$. Clearly, the choice $x_0^i$ leads to the same expressions as $x_0^i + \ell^i$. More than that, the theory with $x_0^i$ is the same as the theory with $x_0^i + \frac{\ell^i}{k}$, reflecting the $\mathbb{Z}_k^d$ translation symmetry

$$(T^i)^k = 1. \tag{120}$$

However, the operators $T^j$ with the choice $x_0^i$ are not the same as with the choice $x_0^i + \frac{\ell^i}{k}$. This is consistent with the extension of the algebra

$$T^i T^j = T^j T^i e^{\frac{2\pi i}{k}\mathcal{Q}_{ij}}, \qquad \mathcal{Q}_{ij} = \mathcal{W}_i^1\mathcal{W}_j^2 - \mathcal{W}_j^1\mathcal{W}_i^1, \tag{121}$$

which can be checked explicitly using the commutation relations (111).

In conclusion, we saw that in the quantum mechanics problem in Section 3.3.2, instantons, i.e., configurations with nonzero $\mathcal{W}_\tau^r$, break the global internal symmetry and lead to its projective representation. Here we see that instantons, i.e., configurations with nonzero $\mathcal{Q}_{i\tau}$ break the translation symmetry and extend it.

Finally, let us discuss the internal symmetry that shifts $\phi^r$. In the quantum mechanics problem in Section 3.3.2, we set the Hamiltonian to zero and we had a $\mathbb{Z}_k^{(1)} \times \mathbb{Z}_k^{(2)}$ symmetry generated by (84). Now, we consider also a nonzero Hamiltonian density $\mathcal{H}$ and it determines the symmetry. Let us assume that $\mathcal{H}$ preserves the full classical $U(1)^{(1)} \times U(1)^{(2)}$ symmetry. Then, it is clear that in the quantum theory we have a $\mathbb{Z}_k^{(1)} \times \mathbb{Z}_k^{(2)}$ symmetry generated by

$$U_r = \exp\left(\frac{i}{V}\int d^d x\,\phi^r + 2\pi i\sum_i\mathcal{W}_i^r\frac{x_0^i - x_*^i}{\ell^i}\right)$$
$$= \exp\left(\frac{i}{V}\int d^d x\left(\phi^r + \sum_i(x_0^i - x_*^i)\partial_i\phi^r\right)\right). \tag{122}$$

The dependence on $x_*^i$ and $x_0^i$ arises as in the discussion around (116). The symmetry operators $U_r$ are independent of $x_*^i$, but they depend on the reference point $x_0^i$.

With an appropriate phase choice of $U_r$, we have

$$U_r^k = 1, \qquad U_r e^{i\phi^s}(U_r)^{-1} = e^{i\phi^s}e^{\frac{2\pi i}{k}\epsilon^{rs}}, \qquad \epsilon^{12} = -\epsilon^{21} = 1, \qquad U_1 U_2 = e^{\frac{2\pi i}{k}}U_2 U_1. \tag{123}$$

The dependence on $x_0^i$ means that these symmetry operators do not commute with translations

$$T^i U_r = U_r T^i e^{\frac{2\pi i}{k}\mathcal{W}_i^r}, \tag{124}$$

and hence, the symmetry algebra of $U_r$ and $T^i$ is extended by the winding symmetry.

---

[15]Recall, $x_*^\mu$ was introduced as a choice of local trivialization. The added terms cancel the dependence on this choice. Instead, they depend on a chosen reference point $x_0^i$.

### 3.4.3 Fuzzy $S^2$

We now use our general prescription in Sections 3.2 and 3.4.1 to lift the quantum mechanics with $\mathcal{P} = S^2$ in Section 3.3.3 to field theory.

Using (63)

$$\mathcal{A} = \frac{i(\bar{z}dz - z d\bar{z})}{2(1 + |z|^2)}, \qquad \mathcal{F} = -\frac{i dz \wedge d\bar{z}}{(1 + |z|^2)^2}, \tag{125}$$

the quantum mechanics Lagrangian (90) has a clear lift to field theory

$$\mathcal{L}_{S^2}^{Classical} = \frac{ik}{2V} \frac{\bar{z}\partial_t z - z\partial_t \bar{z}}{1 + |z|^2} - \mathcal{H}(\partial_i z, \partial_i \bar{z}), \qquad \mathcal{H} = f \sum_i \frac{|\partial_i z|^2}{(1 + |z|^2)^2}, \tag{126}$$

with some real positive coefficient $f$. Here $\mathcal{H}$ was taken to be the standard $SO(3)$ invariant term, which is determined by the metric on the $S^2$ target space $(ds)^2 = \frac{dz d\bar{z}}{(1+|z|^2)^2}$. This is the well-known continuum Lagrangian of a ferromagnet (see, e.g., [3,58,59]).

As we said, in quantum field theory, this Lagrangian cannot be defined by extending it to a bulk as in (79) and the more careful definition in Section 3.4.1 is needed.[16] This definition includes an added term in the Lagrangian density

$$\frac{k}{V} \sum_{irs} (x_*^i - x_0^i) \mathcal{F}_{rs} \partial_t \phi^r \partial_i \phi^s = -\frac{ik}{V} \sum_i (x_*^i - x_0^i) \frac{\partial_t z \partial_i \bar{z} - \partial_i z \partial_t \bar{z}}{(1 + |z|^2)^2}. \tag{127}$$

As in all our cases, this term represents explicit breaking of the translation symmetry.

Also, the system has $d - 2$-form $U(1)$ global symmetry

$$\mathcal{J}_{jt} = \frac{i}{2\pi} \frac{\partial_j z \partial_t \bar{z} - \partial_t z \partial_j \bar{z}}{(1 + |z|^2)^2}, \qquad \mathcal{J}_{ij} = \frac{i}{2\pi} \frac{\partial_i z \partial_j \bar{z} - \partial_j z \partial_i \bar{z}}{(1 + |z|^2)^2},$$

$$\partial_t \mathcal{J}_{ij} = \partial_j \mathcal{J}_{it} - \partial_i \mathcal{J}_{jt}, \qquad \partial_m \mathcal{J}_{ij} + \partial_j \mathcal{J}_{mi} + \partial_i \mathcal{J}_{jm} = 0, \tag{128}$$

$$\mathcal{Q}_{ij} = \int dx^i dx^j \mathcal{J}_{ij} \in \mathbb{Z},$$

which is known as the Skyrmion symmetry. It extends the $\mathbb{Z}_k^d$ symmetry as in (105)

$$T^i T^j = T^j T^i \exp\left(\frac{2\pi i}{k} \mathcal{Q}_{ij}\right). \tag{129}$$

Finally, let us compare this system with the continuum description of an anti-ferromagnet. There, the first term in (126) is replaced as[17]

$$\frac{ik}{2V} \frac{\bar{z}\partial_t z - z\partial_t \bar{z}}{1 + |z|^2} \rightarrow \frac{|\partial_t z|^2}{(1 + |z|^2)^2}. \tag{130}$$

Since the anti-ferromagnet does not have this first-order term, the Lagrangian density is globally well-defined and the subtleties we have been discussing do not arise. As a result, the continuous translation symmetry is not violated and it is not extended.

Following the discussion in Section 2.2.2, we can start with the anti-ferromagnet theory and derive the ferromagnet theory. The system has the $d - 2$-from Skyrmion symmetry (128). We couple it to a background gauge field with a constant magnetic field, i.e., $\sum_i \partial_i A^i = \frac{2\pi k}{V}$. This leads to the term $\frac{ik}{2V} \frac{\bar{z}\partial_t z - z\partial_t \bar{z}}{1 + |z|^2}$ of the ferromagnetic Lagrangian with all its subtleties.

---

[16]As we commented at the end of Section 3.3.3, in this case, the definition using a $U(1)$ gauge theory is familiar to physicists and is referred to as the $CP^1$ presentation of the model.

[17]As in (70), we could have included such a term in $\mathcal{H}$ of the ferromagnet. But if a single time-derivative term is present, then this second-order term is of higher order and can be neglected at low-energies.

Note also that the anti-ferromagnet has a charge-conjugation symmetry $\mathcal{C} : z \to \bar{z}$, and a time-reversal anti-unitary symmetry $\mathcal{T}$, which are absent in the ferromagnet. However, their combination $\mathcal{CT}$ is a time-reversal symmetry of the ferromagnet. Below, we will return to this fact.

## 4 Noninvertible continuous translations

In this section, we will imitate the discussion in [25–28] and show that at least in one of our examples, the one based on $\mathbb{T}^2$ (110), the continuous $U(1)^d$ translation symmetry of the classical theory, which was broken in the quantum theory to a discrete group, is resurrected as a continuous noninvertible translation symmetry.

As mentioned around (113), the relation

$$\mathcal{Q}_{\mu\nu} = \mathcal{W}_\mu^1 \mathcal{W}_\nu^2 - \mathcal{W}_\mu^2 \mathcal{W}_\nu^1 \tag{131}$$

means that the $d-2$-form symmetry associated with $\mathcal{Q}_{\mu\nu}$ follows from the $d-1$-form winding symmetry associated with $\mathcal{W}_\mu^r$. This fact allows us to control the instantons using the $d-1$-form charges $\mathcal{W}_\mu^r$ and thus eliminate their effects including the breaking of the $U(1)^d$ translation symmetry.

Consider the subspace of the Hilbert space with $\mathcal{W}_i^r = 0$ and a projector to this subspace $\mathbf{P}$. (Such projectors were studied in [60,61].) Instantons do not contribute to matrix elements of operators with vanishing winding between states in that subspace. Therefore, in such correlation functions, the continuous translation symmetry is not broken. Another way to see that is that for $\mathcal{W}_i^r = 0$ the dependence on $x_0^i$ in (117) vanishes.

As a result, combining the continuous translation operators with the projector $\mathbf{P}$ removes the instantons that broke the symmetry and the symmetry is resurrected, albeit as a noninvertible symmetry. (It is noninvertible because of the presence of the projector $\mathbf{P}$.)

As a check, the projection on $\mathcal{W}_i^r = 0$, sets also all the $d-2$-form charges $\mathcal{Q}_{ij}$ to zero. This removes the extension of the discrete translation (105), which would not have been consistent with the noninvertible continuous symmetry.

This is very similar to the discussion in [25–28]. In all these cases one starts with a model with a certain global symmetry $G_0$ and gauges a subgroup of it. This gauging breaks part of $G_0$ through an Adler-Bell-Jackiw-like anomaly. However, the gauge theory has another global symmetry $H$ such that this breaking is absent in the $H$ invariant part of the Hilbert space. Then, a broken symmetry transformation can be combined with a projector $\mathbf{P}$ to the $H$-invariant states, such that the effect of the anomaly is absent. In our case, the broken symmetry is the $U(1)^d$ translation symmetry and $H$ is the two $U(1)$ $d-1$-form winding symmetries.

Clearly, this mechanism does not work when there is no global symmetry $H$ with a projector $\mathbf{P}$ that excludes the instantons.

## 5 From the lattice to the continuum

In this section, we discuss lattice models whose low-energy approximations are given by the continuum field theories we studied above. This will give us a better perspective on the discrete translation symmetry of the continuum model.

## 5.1 General discussion

We start with a square lattice in $d$ spatial dimensions with $L^i$ sites in direction $i$ and impose periodic boundary conditions. We denote the total number of sites by

$$\mathcal{N} = \prod_i L^i \,. \tag{132}$$

The relation to the continuum models is obtained by introducing a lattice spacing a such that the physical lengths and the total volume are

$$\ell^i = \mathsf{a}L^i \,, \qquad V = \prod_i \ell^i = \mathsf{a}^d \mathcal{N} \,. \tag{133}$$

At every site, we place a quantum mechanical system with a $U(1)$ gauge symmetry with a Lagrangian term

$$k_{UV} a_t \,, \qquad k_{UV} \in \mathbb{Z} \,. \tag{134}$$

For reasons that will soon be clear, we distinguish between the parameter $k$ of the continuum theories discussed above and this UV value $k_{UV}$. In a Hamiltonian formulation, this means that Gauss law constrains the charge at that site to be $k_{UV}$.[18]

As in Section 3.3, a special case of this corresponds to a theory with a compact phase space $\mathcal{P}$ at each site and and then (134) is replaced with

$$k_{UV} \sum_r \mathcal{A}_r \partial_t \phi^r \,, \qquad k_{UV} \in \mathbb{Z} \,. \tag{135}$$

Clearly, (134) and (135) should be defined more carefully.

For concreteness, let us focus on the case (135). We assume that the interaction between the degrees of freedom at different sites is such that the fields $\phi^r$ at neighboring sites are near each other and therefore we can approximate the system by a continuum field theory with fields $\phi^r$. We emphasize that this assumption is not valid in all the phases of the theory.

Then, the low-energy effective Lagrangian density is the same as (92)

$$\mathcal{L}^{Classical} = \frac{k}{V} \left( \sum_r \mathcal{A}_r(\phi^s) \partial_t \phi^r \right) - \mathcal{H}(\phi^r, \partial_i \phi^r) \,, \qquad k = k_{UV} \mathcal{N} \,. \tag{136}$$

For this description to be valid, $\mathcal{H}$ should includes spatial derivative terms, e.g.,

$$\mathcal{H} = \sum_{rsi} g_{rs}(\phi) \partial_i \phi^r \partial^i \phi^s + \dots \,, \tag{137}$$

with a positive definite metric in field space $g_{rs}$. These terms force the fields $\phi^r$ to be smooth.

As we discussed in the Introduction and around (16), the continuum Lagrangian density (136) is unusual because it depends explicitly on the volume $V$. The lattice construction gives us another perspective about that.

Given a lattice model, it is common to study two distinct limits:

- The thermodynamic limit corresponds to taking the number of sites to infinity, i.e.,

$$L^i \to \infty \,, \quad \text{with fixed} \quad \mathsf{a}, k_{UV} \,. \tag{138}$$

---

[18]If we also discretize Euclidean time, the gauge fields are phases $U_\mu$ on the links. Then, (134) leads to a factor in the integrand of the factional integral that is given by a product over the sites of the lattice $\prod (U_\tau)^{k_{UV}}$. Clearly, this term is gauge invariant and does not need "correction terms." The issues that we have been addressing arise only in the continuum version of this expression.

In this limit, the volume $V$ and $k$ diverge

$$V = \mathsf{a}^d \prod_i L^i \to \infty\,, \qquad k = k_{UV} \prod_i L^i \to \infty\,, \tag{139}$$

such that coefficient of the first term in the Lagrangian density $\frac{k}{V}$ is finite. The fact that $k$ diverges is quite singular. For example, in the $\mathcal{P} = S^2$ model, it means that the spin of the ground state is infinite.

- The continuum limit corresponds to taking the lattice spacing to zero

$$\mathsf{a} \to 0\,, \qquad L^i \to \infty\,, \quad \text{with fixed} \quad \ell^i, k_{UV}\,. \tag{140}$$

In this limit, $k = k_{UV} \prod_i L^i$ diverges and therefore, also the coefficient $\frac{k}{V}$ of the first term in the Lagrangian density diverges. Clearly, this limit is even more singular than the thermodynamic limit.

We see that these two limits are different and both are singular. (A similar difference between the thermodynamic limit and the continuum limit was discussed in a different context in [62].)

Instead of taking such limits, the continuum theory analyzed in this note corresponds to taking $L^i$ large, but finite and then focusing on the low-energy dynamics. It is captured approximately by the continuum Lagrangian (136) with finite $\ell^i$ (and hence, finite $V = \prod_i \ell^i$) and finite $k$.

Now, we can relate the translation symmetry of the lattice model with that of the continuum model. Let $T^i_{UV}$ be the lattice translation operator along direction $i$ by one lattice spacing. In continuum terms, this is translation by

$$\mathsf{a} = \frac{\ell^i}{L^i}\,. \tag{141}$$

Recalling that the continuum translation operator $T^i$ translates by $\frac{\ell^i}{k}$, the lattice translation operator $T^i_{UV}$ is mapped to the continuum operator

$$T^i_{UV} \to (T^i)^{\frac{k}{L^i}}\,, \tag{142}$$

such that

$$(T^i_{UV})^{L^i} \to (T^i)^k = 1 \tag{143}$$

(we do not write an equal sign because the operator in the left-hand-side is a lattice operator, which is represented in the continuum theory by the continuum operator in the right-hand-side).

As a check, the exponent in (142) is an integer

$$\frac{k}{L^i} = k_{UV} \prod_{j \neq i} L^i \in \mathbb{Z}\,. \tag{144}$$

For $d = 1$, it is $k_{UV}$, which is an integer of order one. But for $d > 1$ and large $L^i$, it is a large integer. This means that while the continuum theory does not have continuous translation symmetries, the continuum operators $T^j$ generate translations by much smaller steps than the underlying lattice spacing.

The lattice translation symmetry (143) is clearly Abelian

$$T^i_{UV} T^j_{UV} = T^j_{UV} T^i_{UV}\,. \tag{145}$$

Not only isn't it extended, in fact, the lattice model does not even have the $d-2$-form symmetry associated with $\mathcal{Q}_{ij}$. Interestingly, this is compatible with our picture about the non-Abelian translation symmetry of the continuum theory (105) because

$$
\begin{aligned}
T_{UV}^i T_{UV}^j &\to (T^i)^{\frac{k}{L^i}}(T^j)^{\frac{k}{L^j}} = (T^j)^{\frac{k}{L^j}}(T^i)^{\frac{k}{L^i}}\exp\left(\frac{2\pi i k}{L^i L^j}\mathcal{Q}_{ij}\right) = (T^j)^{\frac{k}{L^j}}(T^i)^{\frac{k}{L^i}}\,, \\
T_{UV}^j T_{UV}^i &\to (T^j)^{\frac{k}{L^j}}(T^i)^{\frac{k}{L^i}}\,,
\end{aligned}
\tag{146}
$$

where we used the fact that $\frac{k}{L^i L^j}$ is an integer. This point is related to the comment around (43) about Abelian subgroups of the extended continuum translation symmetry.

In Sections 5.2 and 5.3, we will present additional issues that are specific to our two examples, $\mathbb{T}^2$ and $S^2$ respectively.

## 5.2  Fuzzy $\mathbb{T}^2$

As we reviewed in Section 3.3.2, in this case, the classical Lagrangian at every site has a $U(1)^{(1)} \times U(1)^{(2)}$ global symmetry, but the quantum theory at every site has only a discrete $\mathbb{Z}_{k_{UV}}^{(1)} \times \mathbb{Z}_{k_{UV}}^{(2)}$ symmetry. Given this fact, there are several natural options for the global symmetry of the Hamiltonian $\mathcal{H}$ in (136).

If $\mathcal{H}$ preserves the classical $U(1)^{(1)} \times U(1)^{(2)}$ symmetry, then we can repeat the discussion around (122)-(124) and find the symmetry of the continuum theory generated by $U_r$.

If $\mathcal{H}$ preserves only the quantum UV symmetry $\mathbb{Z}_{k_{UV}}^{(1)} \times \mathbb{Z}_{k_{UV}}^{(2)}$, the symmetry operators are product of the local symmetry operators of (84) over the lattice $(U_r)_{UV} = \prod e^{i\phi^r}$. They correspond to the continuum operators

$$
(U_r)_{UV} = \prod e^{i\phi^r} \to \exp\left(\frac{i\mathcal{N}}{V}\int d^d x\, \phi^r + 2\pi i\mathcal{N}\sum_i \mathcal{W}_i^r \frac{x_0^i - x_*^i}{\ell^i}\right) = U_r^{\mathcal{N}}\,.
\tag{147}
$$

As a check, (123) and (124) lead to

$$
\begin{aligned}
(U_r^{\mathcal{N}})^{k_{UV}} &= 1\,, \qquad U_r^{\mathcal{N}} e^{i\phi^s}(U_r^{\mathcal{N}})^{-1} = e^{i\phi^s} e^{\frac{2\pi i}{k_{UV}}\epsilon^{rs}}\,, \\
U_1^{\mathcal{N}} U_2^{\mathcal{N}} &= e^{\frac{2\pi i\mathcal{N}}{k_{UV}}} U_2^{\mathcal{N}} U_1^{\mathcal{N}}\,, \qquad (T^i)^{\frac{k}{L^i}} U_r^{\mathcal{N}} = U_r^{\mathcal{N}}(T^i)^{\frac{k}{L^i}}\,,
\end{aligned}
\tag{148}
$$

which are consistent with the lattice symmetry. In particular, this symmetry does not involve the winding charges $\mathcal{W}_i^r$, which are not present on the lattice. Note that the projective phase $e^{\frac{2\pi i\mathcal{N}}{k_{UV}}}$ depends on the number of sites $\mathcal{N}$ modulo $k_{UV}$. We will return to this point in Section 6.3.1.

## 5.3  Fuzzy $S^2$

In this case, the internal symmetry is $SO(3)$ and we study the model in its ferromagnetic phase.

Focusing on the zero modes of the fields, we see that the ground state has spin $\frac{k}{2} = \frac{k_{UV}\mathcal{N}}{2}$. This is consistent with the underlying lattice model, where all the spins are aligned. This fact demonstrates the subtleties in the various limits we discussed in the Introduction and in Sections 2.2 and 5.1. Also, for odd $k$ (which is possible only when both $k_{UV}$ and $\mathcal{N}$ are odd), the global symmetry of the model is realized projectively. We will return to this point in Section 6.3.1.

Of course, this model is extremely well-known and well-studied and it is used to describe ferromagnets. The novelty here is the careful definition of its continuum low-energy theory and its symmetries.

# 6 Conclusions

## 6.1 Summary

We have studied $U(1)$ gauge theories with a classical Lagrangian density and classical action of the form

$$\mathcal{L}_{U(1)}^{Classical} = \frac{k}{V}a_t + \dots, \qquad \mathcal{S}_{U(1)}^{Classical} = \int d^d x \, dt \, \mathcal{L}_{U(1)}^{Classical}. \tag{149}$$

Many systems of interest, including gauge theories with constant charge density and theories with a local compact phase space $\mathcal{P}$ can be presented in this way.

Surprisingly, except in quantum mechanics, i.e., for $d = 0$, the action based on (149) is not meaningful. To make it explicit, we used a trivialization with transition functions at $x_*^i$ and (149) turns out to depend on $x_*^i$. That dependence can be removed by choosing a reference point $x_0^i$ and shifting (149)

$$\frac{k}{V}a_t \rightarrow \frac{k}{V}\left(a_t - \sum_i (x_*^i - x_0^i)f_{it}\right). \tag{150}$$

As a result, the quantum theory is not invariant under continuous translations. The classical $U(1)$ translation symmetry in each direction is explicitly broken

$$U(1) \rightarrow \mathbb{Z}_k, \tag{151}$$

with the remaining translation symmetry generated by $T^i$,

$$(T^i)^k = 1. \tag{152}$$

We interpreted this breaking as a new anomaly due the dynamical $U(1)$ gauge field. In particular, $U(1)$ instantons, i.e., Euclidean space configurations with nonzero $\mathcal{Q}_{i\tau} = \frac{1}{2\pi}\int dx^i d\tau f_{i\tau} \in \mathbb{Z}$, break the translation symmetry.

For $d \geq 2$, the unbroken $\mathbb{Z}_k^d$ translation symmetry is extended by the $d-2$-form magnetic symmetry of the $U(1)$ gauge theory (42)

$$T^i T^j = T^j T^i e^{\frac{2\pi i}{k}\mathcal{Q}_{ij}}, \qquad \mathcal{Q}_{ij} = \frac{1}{2\pi}\int dx^i dx^j f_{ij} = \frac{\ell^i \ell^j}{2\pi V}\int d^d x \, f_{ij}. \tag{153}$$

We then applied this picture to a field theory based on a local phase space $\mathcal{P}$ with coordinates $\phi^r$, Liouville form $\mathcal{A}$, and symplectic form $\mathcal{F} = d\mathcal{A}$. The classical Lagrangian density and classical action are

$$\mathcal{L}_{\mathcal{P}}^{Classical} = \frac{k}{V}\sum_r \mathcal{A}_r \partial_t \phi^r + \dots, \qquad \mathcal{S}_{\mathcal{P}}^{Classical} = \int d^d x \, dt \, \mathcal{L}_{\mathcal{P}}^{Classical}. \tag{154}$$

It is known that when the phase space $\mathcal{P}$ is not simply connected, the definition of the quantum theory is more subtle. We emphasized that the phase space of the field theory in $d \geq 1$ spatial dimensions $\mathcal{P}_d$ is never simply connected.

Then, we expressed this theory as a $U(1)$ gauge theory coupled to a theory whose target space is a circle bundle over $\mathcal{P}$. This allowed us to use the result about the $U(1)$ gauge theory and to write

$$\frac{k}{V}\sum_r \mathcal{A}_r \partial_t \phi^r \rightarrow \frac{k}{V}\left(\sum_r \mathcal{A}_r \partial_t \phi^r - \sum_{irs}(x_*^i - x_0^i)\mathcal{F}_{rs}\partial_i \phi^r \partial_t \phi^s\right) \tag{155}$$

(more correction terms are needed in order to define the Euclidean space action).

We conclude that all these systems have several notions of translations:

- The classical system is invariant under a $U(1)^d$ continuous translation symmetry.

- In the quantum theory, the classical $U(1)^d$ translation symmetry is explicitly broken to $\mathbb{Z}_k^d$. This symmetry is extended as in (153).

- In some cases, the classical $U(1)^d$ translation symmetry can be resurrected as a noninvertible symmetry.

- An underlying lattice model that leads to this continuum theory has an even smaller translation symmetry $\otimes_i \mathbb{Z}_{L^i}$. Its generators $T_{UV}^i$ correspond to the continuum operators $(T^i)^{\frac{k}{L^i}}$. This symmetry is not extended as in (153). (For $d = 1$ and $k_{UV} = 1$, we have $k = L$ and the $\mathbb{Z}_L$ lattice translation symmetry is the same as the $\mathbb{Z}_k$ translation symmetry of the continuum theory.)

## 6.2 Broader perspective

Following the discussion in Section 2.2.2, we can phrase our entire discussion as follows. A more detailed description of this line of thinking will be presented in [63].

We consider a theory with a global $d-2$-form global symmetry

$$\mathcal{J} = \frac{1}{2} \sum_{\mu\nu} \mathcal{J}_{\mu\nu} dx^\mu \wedge dx^\nu, \qquad d\mathcal{J} = 0, \qquad \mathcal{Q} = \int \mathcal{J} \tag{156}$$

(for $d = 1$, this symmetry can be thought of as a $-1$-form symmetry). We couple it to a background $d-1$-form gauge field $A$ through

$$\int \mathcal{J}A. \tag{157}$$

The main subtlety is the precise definition of (157). $\mathcal{J}$ is a well-defined operator. But for topologically nontrivial $A$ some care is needed.

We are interested a background $A$ such that

$$dA = \frac{2\pi k}{V} dx^1 \wedge dx^2 \cdots \wedge dx^d. \tag{158}$$

Such an $A$ must depends explicitly on the coordinates and have nontrivial transition functions. As a result, the continuous translation symmetry is explicitly broken and being extended by $\mathcal{Q}$.

A simple, well-known example of this is a $d = 2$ system on a torus with an ordinary $U(1)$ global symmetry coupled to constant background magnetic field. In this case the continuous translation symmetry is explicitly broken to a discrete symmetry, which is furthermore extended by $\mathcal{Q}$.[19] (Note that in this example, the magnetic field is the background $dA$. It is not the magnetic field of the dynamical field $da$ in the examples in Section 2).

In the examples considered in this paper, the breaking of translation symmetry is less obvious for the following reason. In all these examples, the conservation equation $d\mathcal{J} = 0$ is topological and does not rely on the classical equations of motion. Therefore, we can write locally $\mathcal{J} = d\Phi$, where $\Phi$ is not globally well-defined. Then, we can express (157) as

$$\frac{2\pi k}{V} \int d^d x dt \Phi_t, \tag{159}$$

which seems translation invariant. Indeed, the corresponding classical theory is translation invariant. However, as we explained, the proper definition of the local expression (159) shows that despite appearance, the quantum theory is not translation invariant.

---

[19]On a plane, rather than on a torus, the translation symmetry is continuous.

## 6.3 't Hooft anomalies

We end this note by offering some thoughts about 't Hooft anomalies in our systems and related systems. (Recall the distinction between 't Hooft anomalies and Adler-Bell-Jackiw anomalies, which we summarized in footnote 5.)

Before we start, we should clarify that even though in spacial cases, it is clear what 't Hooft anomaly involving translations means [64], this is not always straightforward. The reason is that it is not obvious how to couple translations to a background gauge field and therefore we cannot formulate the anomaly as an obstruction to gauging.

### 6.3.1 Relation to Lieb-Schultz-Mattis theorem

Our continuum systems have various internal symmetries. The system with a $\mathbb{T}^2$ target space has a discrete symmetry and two $U(1)$ $d-1$-form winding symmetries. And the system with an $S^2$ target space has an $SO(3)$ 0-form symmetry and a $U(1)$ $d-2$-form Skyrmion symmetry. And all our $U(1)$ gauge theories have a $U(1)$ $d-2$-form magnetic symmetry. These symmetries can have their own 't Hooft anomalies as well as mixed anomalies with the spatial symmetries.

Even without studying these anomalies, we can resolves a puzzle that was one of our original motivations for this note.

Consider the lattice models in Section 5 with a quantum mechanical model based on the compact phase space $\mathcal{P}$ at every site. It is often the case that these quantum mechanical systems have a global symmetry $G$, which acts projectively on the local Hilbert space. For example, the $\mathbb{T}^2$ models have a $G = \mathbb{Z}^{(1)}_{k_{UV}} \times \mathbb{Z}^{(2)}_{k_{UV}}$ symmetry, which is realized projectively (except for $k_{UV} = 1$). And the $S^2$ models have a $G = SO(3)$ symmetry, which is realized projectively for odd $k_{UV}$. We assumed that the full lattice model respects this internal global symmetry $G$.

Whenever the symmetry $G$ acts projectively on the local Hilbert space, the lattice model has a mixed 't Hooft anomaly between lattice translations and the internal symmetry $G$ [64–72]. This anomaly leads to the modern version of the celebrated Lieb-Schultz-Mattis theorem [73, 74].

One aspect of this anomaly is that depending on $k_{UV}$ and the total number of sites $\mathcal{N}$, the whole system might be in a projective representation of $G$. In the $\mathbb{T}^2$ model, this happens when $\mathcal{N} \bmod k_{UV} \neq 0$ (see Section 5.2). And in the $S^2$ model, this happens when $k = k_{UV}\mathcal{N}$ is odd (see Section 5.3).

Let us focus on the $S^2$ models (Section 3.4.3). The low-energy continuum field theory in its antiferromagnet phase does not have the first term in (6). Then, this anomaly is matched in the continuum using the fact that a new internal $\mathbb{Z}_2^{\mathcal{C}}$ charge-conjugation symmetry emanates from lattice translation [64,69] and the lattice anomaly becomes an ordinary 't Hooft anomaly between the internal symmetry $G = SO(3)$ of the lattice model and this emanant $\mathbb{Z}_2^{\mathcal{C}}$ symmetry. See also [75, 76].

As we vary the parameters of the lattice model, we can move to a ferromagnetic phase.[20] The lattice symmetry and its 't Hooft anomalies are unchanged. However, as was emphasized at the end of Section 3.4.3, no such $\mathbb{Z}_2^{\mathcal{C}}$ emanant symmetry is present in the ferromagnetic phase of the same system. How can the anomalies match in that case?

Another aspect of this puzzle is that the Lieb-Schultz-Mattis anomaly depends on whether $k_{UV}$ is even or odd, i.e., whether the microscopic spins $s = \frac{k_{UV}}{2}$ are integer or half-integer. In the anti-ferromagnetic case, this distinction is visible also in the IR continuum theory. However, this is not the case in the ferromagnetic phase. In that case, it is clear that the continuum theory depends only on the integer $k = k_{UV}\mathcal{N}$. How does the dependence on $k_{UV}$ appear?

---

[20]Since we work in finite volume, the notion of distinct ferromagnetic or anti-ferromagnetic phases is imprecise. What we mean here is the finite volume system that looks like these phases in the infinite volume limit.

By now, the answer to this question should be obvious. In a ferromagnet, the translation symmetry of the continuum theory is discrete, generated by $T^i$. And the lattice translation generators $T^i_{UV}$ act in the continuum theory as powers of these (142)

$$T^i_{UV} \to (T^i)^{\frac{k}{L^i}}. \tag{160}$$

Since the continuum theory has no continuous translation symmetry, there is no reason why the discrete translation symmetry $T^i$ does not have anomalies. Indeed, we expect it to be such that $(T^i)^{\frac{k}{L^i}}$ has the same 't Hooft anomaly with the internal symmetry $G$ as $T^i_{UV}$ has on the lattice.

Let us demonstrate it in a simple case. For $d = 1$, the long-distance behavior of the Heisenberg model in its anti-ferromagnetic phase is the same as that of the $O(3)$ sigma-model with $\theta = \pi k_{UV}$. It is gapless for odd $k_{UV}$ and gapped for even $k_{UV}$. The lattice anomaly is present only for odd $k_{UV}$ and then it is matched with the anomaly in the low-energy field theory. In the ferromagnetic phase, the low-energy theory is gapless and it is described by $\mathcal{L}_{S^2}$ for all $k_{UV}$. It depends only on the integer $k = k_{UV}L$. Its $\mathbb{Z}_k$ translation symmetry is generated by $T$ and we expect this $T$ to have a mixed 't Hooft anomaly with the $SO(3)$ symmetry. The underlying lattice model has a $\mathbb{Z}_L \subseteq \mathbb{Z}_k$ translation symmetry, generated by $T_{UV}$, which corresponds to the continuum operator $T^{k_{UV}}$. The fact that $T_{UV}$ has a mixed anomaly with $SO(3)$ only for odd $k_{UV}$ is matched with the continuum anomaly using the identification $T_{UV} \to T^{k_{UV}}$.

### 6.3.2 Relation to filling constraints

In our gauge theory systems, the $U(1)$ charge density was constrained locally to be $\frac{k}{V}$, leading to our new anomaly. This is to be contrasted with a system with a global $U(1)$ symmetry with a chemical potential. The microcanonical presentation of this system involves a constraint on the total $U(1)$ charge. As explained in [64], in this case, there is an 't Hooft anomaly between the $U(1)$ global symmetry and translations. This anomaly underlies Oshikawa's presentation [74] of Luttinger's theorems and other filling constraints. It is tempting to suggest that upon coupling the systems with a global $U(1)$ symmetry to dynamical gauge fields with the coupling (1), this 't Hooft anomaly leads to the Adler-Bell-Jackiw-like anomaly we discussed in this note.

## Acknowledgments

We are grateful to Ofer Aharony, Tom Banks, Dan Freed, Helmut Hofer, Juan Maldacena, Max Metlitski, Gregory Moore, Abhinav Prem, Shinsei Ryu, Sahand Seifnashri, Senthil Todadri, Shu-Heng Shao, Steve Shenker, Wilbur Shirley, Dam Son, Nikita Sopenko, Yuji Tachikawa, Ashvin Vishwanath, and Edward Witten for useful discussions.

**Funding information** This work was supported in part by DOE grant DE-SC0009988 and by the Simons Collaboration on Ultra-Quantum Matter, which is a grant from the Simons Foundation (651444, NS).

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
