# Peer review of "Ferromagnets, a New Anomaly, Instantons, and (noninvertible) Continuous Translations"

_SciPost Physics, doi:SciPost Phys. 18, 063 (2025)_

## Round 3 · Referee Report · Anonymous (Referee 1) · 2025-1-11

Strengths

This paper studies the fate of translational symmetry in field theories with a dynamical U(1) gauge field at finite charge density. It is explained in detail why such theories, despite having continuous translation symmetry at the classical level, break this symmetry to a discrete subgroup after quantization. The physics is deeply related to the presence of a 't Hooft anomaly between U(1) and translations when the U(1) gauge field is treated as a background field; gauging the U(1) then leads to an ABJ anomaly. In some cases there is also a non-invertible continuous translation symmetry.

The paper is extremely clearly written and tackles fundamental questions for quantum field theories that are relevant in condensed matter physics. Some old observations in the literature are explained in a new and modern way from a more general perspective.

Weaknesses

It is not clear if the results of the paper imply new physical predictions for condensed matter systems. Are there some observational consequences for Fermi liquids or superfluids where there is no underlying lattice?

Report

The journal's acceptance criteria are met and I recommend publication in SciPost.

Requested changes

  1. The comment at the top of page 46 is somewhat confusing:

"A simple, well-known example of this is a d = 2 system with an ordinary U(1) global symmetry coupled to constant background magnetic field. In this case the continuous translation symmetry is explicitly broken to a discrete symmetry, which is furthermore extended by Q. "

Quantum Hall systems, such as an integer filled Landau level in the continuum, can preserve a continuous magnetic translation symmetry. It would be helpful if the author could clarify the comment in light of this example.

  1. The author credits Ref. 64 - 71 for the observation that the Lieb-Schulz-Mattis filling constraints are related to a mixed 't Hooft anomaly between U(1) and translation symmetry. The earliest paper cited is by Thorngren and Else, Ref. 65. I believe this observation was first presented in https://arxiv.org/abs/1511.02263 , albeit with somewhat different wording, and formed the basis for the later references.

Recommendation

Publish (surpasses expectations and criteria for this Journal; among top 10%)

  • validity: top
  • significance: top
  • originality: top
  • clarity: top
  • formatting: perfect
  • grammar: perfect

Author:  Nathan Seiberg  on 2025-01-31  [id 5175]

(in reply to Report 1 on 2025-01-11)

Dear Editor,

I'd like to thank the referees for their thoughtful comments.

Unfortunately, it still too early to know whether "the results of the paper imply new physical predictions for condensed matter systems." Clearly, it will be wonderful if such implications are found.

The comment at the top of page 46 applies to the system on a spatial torus. This will be clarified in the revised version.

I apologize for the omission of the reference. This will be fixed in the revised version.

Sincerely yours, Nathan Seiberg

---

## Round 3 · Referee Report · Anonymous (Referee 2) · 2025-1-31

Report

This submission has been delayed for a long time and the other two referees who have accepted to review the paper keep unresponsive for several months. I have briefly went through the paper and I agree with the first referee that, this paper is clearly written and addresses fundamental problems at the intersection between quantum field theories and condensed matter physics. I thintk there is no need to further delay the publication of this paper.

The editor in charge

Recommendation

Publish (surpasses expectations and criteria for this Journal; among top 10%)

---

## Round 3 · Referee Report · Anonymous (Referee 3) · 2025-2-16

Strengths

1- Identifies and elucidates a very subtle point in a very broad class of systems. 2- Presents both general and explicit examples of the ABJ anomaly on different manifolds, and a connection to Landau level physics.

Weaknesses

1- It is hard to tell what is the operational consequence of this translation breaking. I wonder where in an experiment at finite charge density do we end up making a choice of origin?

Report

Overall, I quite enjoyed this paper. It presents a very fundamental and subtle puzzle for those of us who think about gauge theories describing condensed matter systems, and offers a satisfying and careful analysis of this puzzle. It certainly gave me a lot to think about, and I learned new things from it.

I cannot really fault the paper for leaving many interesting questions open, in fact, I would say that is a feature, and so the paper meets at least the acceptance criterion 2 for publication in SciPost physics.

Recommendation

Publish (meets expectations and criteria for this Journal)

---

## Editorial Decision

published